# 2D vanadium carbide MXenzyme to alleviate ROS-mediated inflammatory and neurodegenerative diseases

Wei Feng [1,2], Xiuguo Han [3], Hui Hu[4,5], Meiqi Chang[2], Li Ding[2], Huijing Xiang[1], Yu Chen [1,2✉] & Yuehua Li[4✉]

Reactive oxygen species (ROS) are generated and consumed in living organism for normal metabolism. Paradoxically, the overproduction and/or mismanagement of ROS have been involved in pathogenesis and progression of various human diseases. Here, we reported a two-dimensional (2D) vanadium carbide ($V_2C$) MXene nanoenzyme (MXenzyme) that can mimic up to six naturally-occurring enzymes, including superoxide dismutase (SOD), catalase (CAT), peroxidase (POD), glutathione peroxidase (GPx), thiol peroxidase (TPx) and halo-peroxidase (HPO). Based on these enzyme-mimicking properties, the constructed 2D $V_2C$ MXenzyme not only possesses high biocompatibility but also exhibits robust in vitro cyto-protection against oxidative stress. Importantly, 2D $V_2C$ MXenzyme rebuilds the redox homeostasis without perturbing the endogenous antioxidant status and relieves ROS-induced damage with benign in vivo therapeutic effects, as demonstrated in both inflammation and neurodegeneration animal models. These findings open an avenue to enable the use of MXenzyme as a remedial nanoplatform to treat ROS-mediated inflammatory and neurode-generative diseases.

[1] School of Life Sciences, Shanghai University, Shanghai, P. R. China. [2] State Key Laboratory of High Performance Ceramics and Superfine Microstructures, Shanghai Institute of Ceramics, Chinese Academy of Sciences, Shanghai, P. R. China. [3] Department of Orthopedic Surgery, Xin Hua Hospital Affiliated to Shanghai Jiao Tong University School of Medicine, Shanghai, P. R. China. [4] Institute of Diagnostic and Interventional Radiology, Shanghai Jiao Tong University Affiliated Sixth People's Hospital, Shanghai, P. R. China. [5] Medmaterial Research Center, Jiangsu University Affiliated People's Hospital, Zhenjiang, China. ✉email: chenyuedu@shu.edu.cn; liyuehua312@163.com

Reactive oxygen species (ROS) are chemically reactive molecules containing oxygen, including singlet oxygen ($^1O_2$), superoxide anion radical ($O_2^{-}\bullet$), hydroxyl radical ($\bullet OH$), and hydrogen peroxide ($H_2O_2$)[1]. At low concentrations, ROS play an essential role in adjusting cell functions[2]. However, mismanagement and/or overproduction of ROS would subject a biosystem to cause oxidative stress. Excess ROS result in irreversible oxidative damage to the biomacromolecules (e.g., lipids, nucleic acids, and proteins), induce a variety of deleterious cellular responses (e.g., apoptosis and necrosis), and trigger a myriad of pathologies (e.g., atherosclerosis, neurodegeneration, inflammation, aging, hemochromatosis, and even cancer)[3,4]. Therefore, to remit its detrimental effect, tight ROS regulation is crucial for maintaining cellular homeostasis.

Under normal circumstance, the intracellular redox equilibrium to resist oxidative stress is sustained by a collection of enzymatic antioxidants, primarily consisting of superoxide dismutase (SOD), catalase (CAT), peroxidase (POD), thiol peroxidase (TPx), glutathione peroxide (GPx), etc.[5–7]. However, natural enzymes generally involve proteins and RNA molecules, which are susceptible to environmental factors and become inactive under pathological conditions. To circumvent these drawbacks, artificial enzymes termed as "nanozyme" with intrinsic enzyme-like characteristics have been designed and constructed as alternatives to exert enzyme functionality. Owing to their exceptional properties (facile preparation, high stability, tunable activity, etc.), nanozymes garner great attention and enable a broad spectrum of biomedical applications. Recently, exciting paradigms can be found in the emerging field of nanozymes, such as iron-based nanoparticles[8–10], copper-based nanoreactors[6], carbon-based nanoplatforms[11,12], nanoceria[13], noble metal-based nanomaterials (e.g., Au, Pt, and Pd)[14–17], organic nanoenzymes[18–20], etc. Thereinto, vanadium, one of the 40 essential micronutrients, possesses a regulatory role in the biological system[21,22]. As a consequence of the intrinsic catalytic activities toward classical peroxidase substrates and long-term anti-biofouling abilities, vanadium-based nanomaterials have increasingly aroused considerable interest in the field of nanozymes[23,24]. Despite the high prospect, to the best of our knowledge, very few vanadium-based nanozymes have been developed so far[24,25]. These nanozymes, in particular, mainly concentrate on the single enzymic function in vitro and almost neglect the in vivo effect. More importantly, an artificial enzyme simply with a single function is incapable of mimicking the natural intracellular antioxidant system in combating oxidative stress. The current trend in nanozyme development has focused on constructing a nanoreactor to simulate the sophisticated intracellular enzyme-participated ROS defense system. Accordingly, it is necessary to develop more robust vanadium-based nanomaterials that can functionally mimic the sophisticated cellular antioxidant enzymes. Recent advances in transition metal carbides and/or nitrides (MXenes) suggest that MXenes have been regarded as a thriving class of two-dimensional (2D) materials in numerous potential applications ranging from energy storage to biomedicine, attributed to their versatile compositions, physicochemical diversity, and tailorability[26–29]. Even though some impressive preliminary results have been achieved in theranostics, biosensing, bioimaging, and antibacterial, the exploration of other scenarios of MXenes in vitro and in vivo are still urgently required.

In this study, we report 2D vanadium carbide ($V_2C$) MXene as a successful paradigm of guiding nanozymes to implement antioxidative behaviors for ROS elimination under pathophysiological conditions. The interdependent relationship between MXene and enzyme inspires us to propose the concept of "MXenzyme": MXenzyme is in analogy to the nomenclature of nanozyme, highlighting enzyme-mimicking characteristics of the MXenes. We make the interesting discovery that 2D $V_2C$ MXenzyme is a kind of artificial nanozyme conducting intrinsic multiple enzyme-like

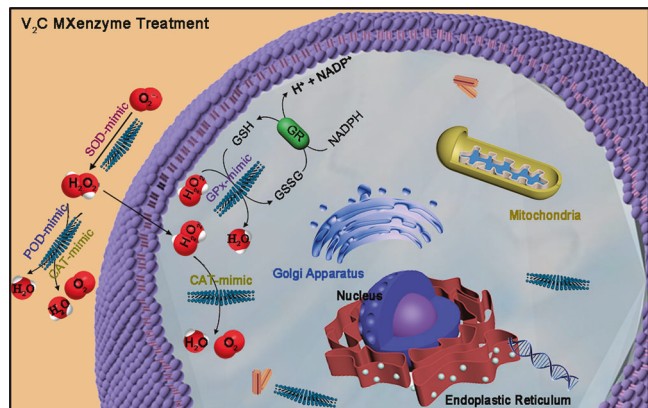

**Fig. 1 Schematic illustration of ROS-scavenging activities of $V_2C$ MXenzyme with multiple enzyme-mimicking properties.** $V_2C$ MXenzyme effectively catalyzes $O_2^{-}\bullet$ into $H_2O_2$ and $O_2$, decomposes $H_2O_2$ into $O_2$ and $H_2O$, and gets rid of $\bullet OH$.

activities, including SOD, CAT, POD, TPx, GPx, and haloperoxidase (HPO), which are in a position to catalyze ROS. Our results demonstrate that 2D $V_2C$ MXenzyme effectively catalyzes $O_2^{-}\bullet$ into $H_2O_2$ and $O_2$, decomposes $H_2O_2$ into $O_2$ and $H_2O$ and gets rid of $\bullet OH$, as well as restrains ROS elevation by in vivo tests. Intelligent cytoprotection against oxidative stress-induced inflammation and neurotoxicity can be achieved by the concerted catalysis of multifunctional MXenzymes (Fig. 1, Supplementary Figs. 1 and 2). This strategy not only sheds light on a type of nanozymes with multiple enzyme-mimicking properties and excellent ROS-removal efficacy but also paves an avenue toward broadening the bioapplications of MXenes into catalytic nanomedicine.

## Results and discussion

**Synthesis and characterization of 2D $V_2C$ MXene**. 2D $V_2C$ MXene was synthesized via a facile exfoliation and intercalation procedure (Fig. 2a)[30]. In a typical synthesis, the interlaced aluminum layers were selectively extracted from the corresponding MAX phase precursors $V_2AlC$ through wet-chemical etching using hydrofluoric acid (HF) solution. Then, the as-etched multilayered (ML) $V_2C$ MXene was delaminated by employing tetrapropylammonium hydroxide (TPAOH) as an intercalant to reduce the interaction between individual layers and subsequently obtain few-layered (FL) $V_2C$ MXene. As visualized by field-emission scanning electron microscopy (FESEM) images, pristine $V_2AlC$ exhibits closely compacted layered platelet morphology (Fig. 2b). After HF treatment, the compact layers expanded and became loose accordion-shape ML structure (Fig. 2c), demonstrating the successful exfoliation. The energy dispersive X-ray (EDX) element mapping (Supplementary Fig. 3a), and corresponding elemental analysis (Supplementary Fig. 3b and Supplementary Table 1) confirm the uniform distribution of V, Al, and C elements in MAX phase ceramic. In addition, a trimodal configuration of V, Al, and C in the hierarchical nanoflake structure is further revealed by linear-scanning EDX spectra along the lateral-section of the sample (Supplementary Fig. 4), which shows that the signal areas of V and C are similar while the signal pattern of Al appears opposite to V and/or C.

TEM images and corresponding selected area electron diffraction (SAED) pattern provide evidence that $V_2AlC$ is featured with a laminar hexagonal crystal structure with a space group of P6$_3$/mmc (Fig. 3a–c). In order to investigate more details with the crystallographic structure and chemical composition, the atomic-resolution scanning transmission electron microscopy (STEM) in high-angle annular dark-field (HAADF) mode was performed,

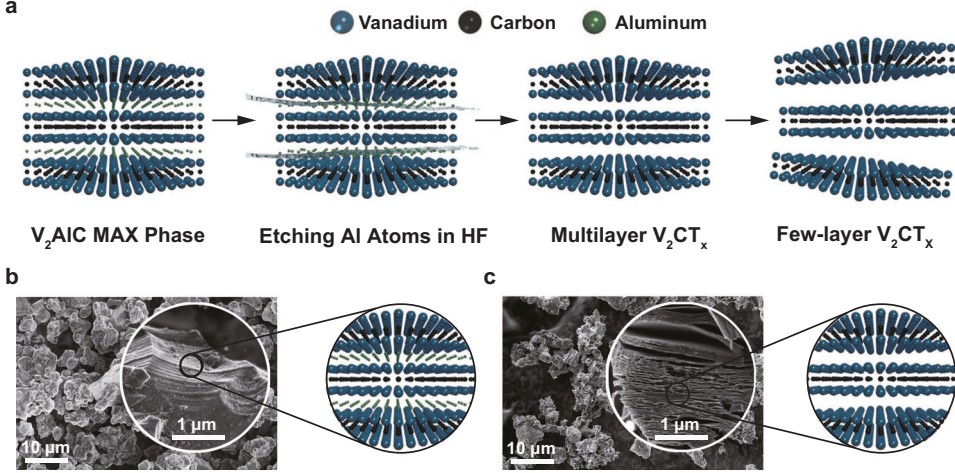

**Fig. 2 Synthesis and characterization of V₂AlC MAX and V₂C MXene. a** Schematic diagram of selective etching and interlayer expansion route to fabricate V₂C MXene from the parent V₂AlC MAX phase. **b** FESEM images (Inset depicts the corresponding high-magnification SEM image) and structural model of V₂AlC before HF treatment, displaying a typical compact layer structure. **c** FESEM images (Inset depicts the corresponding high-magnification SEM image) and structural model of V₂AlC after HF treatment to fabricate multilayer V₂C MXene with a typical exfoliated layer topology. A representative image of three replicates from each group is shown.

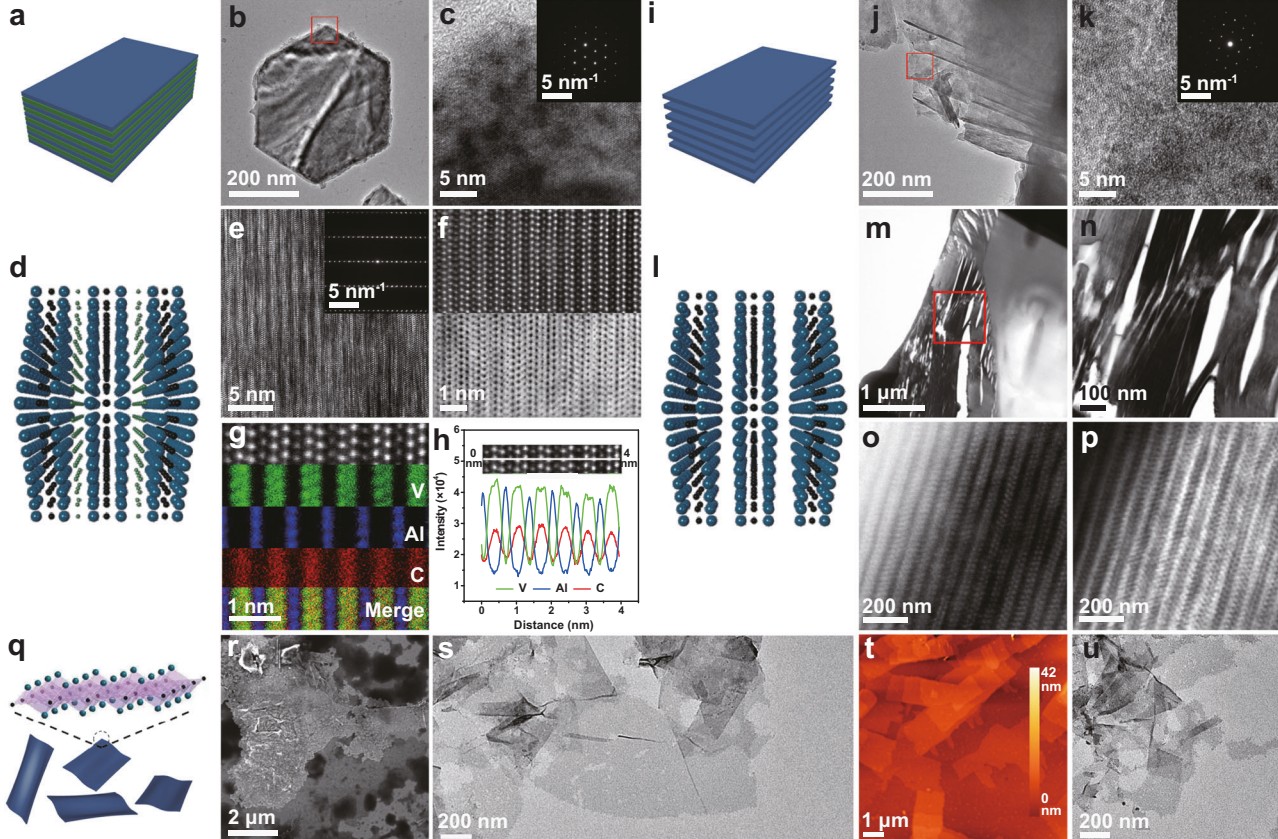

**Fig. 3 Structural and compositional characterizations of V₂AlC MAX and V₂C MXene. a** Schematic of V₂AlC structure. Top view of **b** low and **c** high magnification (Inset describes the corresponding SAED pattern) TEM images of V₂AlC. **d** Atomistic model of layer structure showing in-plane chemical ordering in V₂AlC MAX phase. **e** HRSTEM image (Inset illustrates the corresponding SAED pattern) and **f** enlarged HRSTEM images (upper presents HAADF image and lower presents ABF image) of V₂AlC. **g** HRSTEM with corresponding EDX mapping of V, Al, and C element signals of V₂AlC along [001] zone axis. **h** EDX linear-scanning profile of V, Al, and C element over the white line marked in the HAADF image. **i** Schematic of ML V₂C MXene. Top view of **j** low and **k** high magnification (inset exhibits the corresponding SAED pattern) TEM images of ML V₂C MXene. **l** Atomistic model of layer structure elucidating in-plane chemical ordering in ML V₂C MXene. **m, n** Overview cross-sectional HRSTEM micrograph of ML V₂C MXene after HF selective etching of Al. **o** HAADF and **p** ABF image of ML V₂C MXene. **q** Schematic of FL V₂C MXene. **r** FESEM, **s** TEM, and **t** AFM image of FL V₂C MXene after intercalation with TPAOH. **u** TEM image of FL V₂C MXene after sonication. A representative image of three replicates from each group is shown.

which is sensitive to the element atomic number. It is worthwhile to mention that the larger atomic number V ($Z = 23$) elements scatter the electrons more strongly than Al ($Z = 13$) elements, therefore, V atoms appear brighter contrast in this HAADF images[31]. Cross-sectional HRSTEM images of $V_2AlC$ (Fig. 3e, f), as obtained along the [001] zone axis, signify a purity phase and highly-ordered lamellar arrangement with alternating dark and bright contrast, as schematically depicted in Fig. 3d. Furthermore, the corresponding EDX elemental mapping of V, Al, and C atoms in the sample reveals that the layer stacking order, excluding C, is V–V–Al–V–V, where Al layer is sandwiched between two $V_2C$ layers (Fig. 3g). As illustrated in the EDX linear-scanning analysis, the linear profiles imply a similar trend in variation of V and C, whereas the dips for V and/or C are corresponding to the peaks of Al and vice versa, undoubtedly proving that Al is interleaved into two $V_2C$ (Fig. 3h). In addition, the distance between two neighboring V layers is about 1.85 Å, the interval between V and Al is around 2.60 Å, and the separation between two adjacent Al layers is approximately 6.65 Å.

After selectively etching the Al layers, the obtained ML $V_2C$ MXene presents the regular and homogenous layered structure with hexagonal lattice features (Fig. 3i–k). It is noted that in the cross-sectional overview micrograph (Fig. 3l–n), after treating with HF to remove Al from the densely packed MAX phase, the cavities form, which is several to dozens of nanometers wide and distributes periodically, indicates that the layers are split from each other. Careful inspection of atomic-resolution STEM images (Fig. 3o, p), EDX element mapping (Supplementary Figs. 5a and 6a) and corresponding elemental analysis (Supplementary Figs. 5b, 6b, and Supplementary Table 2) prove that almost no Al layer is remaining and the region is fully transformed into MXene. Notably, after TPAOH delamination, the exfoliated $V_2C$ MXene nanoflakes with a lateral size of several micrometers are thin enough nearly to be electron-transparent (Fig. 3q–s), which theoretically is a three-atom-thick material consisting of two layers of V and one layer of C. As revealed in the atomic force microscopy (AFM) measurements (Fig. 3t, Supplementary Fig. 7), the obtained MXene possesses a relatively uniform thickness of around 2.7 nm, a little thicker than the theoretical thickness, demonstrating that few layers of MXene were obtained, and the trapped water and other molecules could also contribute to the total thickness[28]. The STEM images with EDS mapping results further confirm that Al element has been selectively etched, and V, C, F, and O atoms are uniformly distributed throughout the entire nanoflakes, where F and O atoms originate from the introduced groups of –F, –O, and –OH (Supplementary Fig. 8, Supplementary Table 3). Finally, small and ultrathin 2D $V_2C$ MXene nanoflakes could be readily acquired by sonication (Fig. 3u).

**Enzyme-mimicking activities of 2D $V_2C$ MXenzyme**. The multiple enzyme-mimicking properties of 2D $V_2C$ MXenzyme were assessed by selecting the ordinary substrates for natural enzymes under physiological conditions. Because $O_2^{-\bullet}$ is perpetually generated in normal body metabolism, SOD, as a key antioxidant enzyme in cells against ROS, catalyzes the dismutation reaction of $O_2^{-\bullet}$ into $O_2$ and $H_2O_2$, thus SOD mimetics can be employed as potential therapeutic agents against a number of oxidative stress-triggered illnesses[32]. We initially explored the SOD-like activity of $V_2C$ MXenzyme. $O_2^{-\bullet}$ is commonly produced in situ through the reaction between xanthine (Xan) and xanthine oxidase (XOD)[33]. The capability of $V_2C$ MXenzyme to quench $O_2^{-\bullet}$ was investigated using 2-(4-iodophenyl)-3-(4-nitrophenyl)-5-(2,4-disulfophenyl)-2H-tetrazolium) (WST-1), which could work with $O_2^{-\bullet}$ to generate spectrophotometrically

identifiable formazan at 450 nm[6]. It is noteworthy that the amount of formazan markedly decreased with increasing $V_2C$ MXenzyme concentration (Fig. 4a), indicating that $V_2C$ MXenzyme is gifted with effective SOD-like activity.

CAT or CAT mimics have a well-documented capability to catalyze the decomposition of two molecules of $H_2O_2$ for yielding $O_2$ and $H_2O$, therefore preventing the accumulation of $H_2O_2$ and protecting living organisms from oxidative damage by peroxide[32]. Terephthalic acid (TPA), a nonfluorescent probe, reacts with $H_2O_2$ to be converted into a fluorescent indicator 2-hydroxyterephthalic acid with a fluorescence characteristic peak at 425 nm[6]. The fluorescent intensity displays a significant downtrend in the presence of $V_2C$ MXenzyme, demonstrating that $V_2C$ MXenzyme can consume $H_2O_2$ (Fig. 4b, Supplementary Fig. 9). Subsequently, the CAT-like activity of the $V_2C$ MXenzyme was measured by monitoring the change in the absorbance of $H_2O_2$ at 240 nm (Supplementary Fig. 10a)[5]. As expected, there is a distinct reduction in the $H_2O_2$ absorbance as a function of time, suggesting that the $V_2C$ MXenzyme functionally mimics CAT. In addition, the CAT-like activity of $V_2C$ MXenzyme can also be testified by detecting the $H_2O_2$-triggered $O_2$ generation with the assistance of a dissolved-oxygen meter. In the presence of $V_2C$ MXenzyme, $O_2$ bubbles can be observed in the solution (Supplementary Fig. 10b) and the $O_2$ production is increased with the concentration of $H_2O_2$ (Supplementary Fig. 10c), providing the direct evidence that $V_2C$ MXenzyme could act as a kind of efficient CAT mimics to scavenge $H_2O_2$. In order to identify the CAT-like enzymatic catalysis mechanism, a steady-state kinetic assay was carried out by varying the concentration of $H_2O_2$ (2–400 mM) at a fixed concentration of $V_2C$ MXenzyme, which followed the representative Michaelis–Menten kinetics (Supplementary Fig. 10c–e).

POD is acknowledged as an another type of antioxidant enzyme that can detoxify the $H_2O_2$ to $H_2O$[33,34]. In a standard procedure, 3,3′,5,5′-tetramethylbenzidine (TMB) was selected as the chromogenic substrate to assess the POD-like activity. In the presence of TMB, $V_2C$ MXenzyme catalyzes $H_2O_2$ into $H_2O$. Meanwhile, the colorless TMB is converted into blue-colored oxidized TMB (TMBox) with a maximum characteristic absorbance at 652 nm (Supplementary Fig. 11a, b), which is a noteworthy time-dependent augmentation after the addition of the $V_2C$ MXenzyme into $H_2O_2$ aqueous solution (Fig. 4c), while the control groups without $V_2C$ MXenzyme or $H_2O_2$ display inconspicuous color variation (Supplementary Fig. 11a, b). $V_2C$ MXenzyme possesses similar activities to natural POD, i.e., the POD-like activity of $V_2C$ MXene is pH sensitive (Supplementary Fig. 11c). In addition, after a time-scanning mode measurement, the obtained curves followed the typical Michaelis–Menten equation confirming the POD-like behavior of $V_2C$ MXene (Supplementary Fig. 11d–f).

GPx, an enzyme family with POD activity playing a critical role in maintaining $H_2O_2$ level, utilizes cellular tripeptide glutathione (GSH) as a reductant to catalyze the reduction of $H_2O_2$ to $H_2O$ accompanying by the transformation of reduced GSH to oxidized glutathione (GSSG), followed by reduction GSSG to produce GSH with the assistance of GSH reductase (GR) and coenzyme nicotinamide adenine dinucleotide phosphate (NADPH)[5]. Accordingly, the GPx-like activity of $V_2C$ MXenzyme was determined by spectrophotometrically real-time measuring the decrease of NADPH level at 340 nm (Fig. 4d)[5,6]. The absorbance of NADPH is reduced rapidly with time and up-regulated $V_2C$ MXenzyme concentrations (Fig. 4d, Supplementary Fig. 12a), revealing the GPx-mimicking activity of $V_2C$ MXenzyme. The GPx-like catalytic property of $V_2C$ MXenzyme was further estimated by a steady-state kinetic test employing $H_2O_2$ as the substrates (Supplementary Fig. 12b–d).

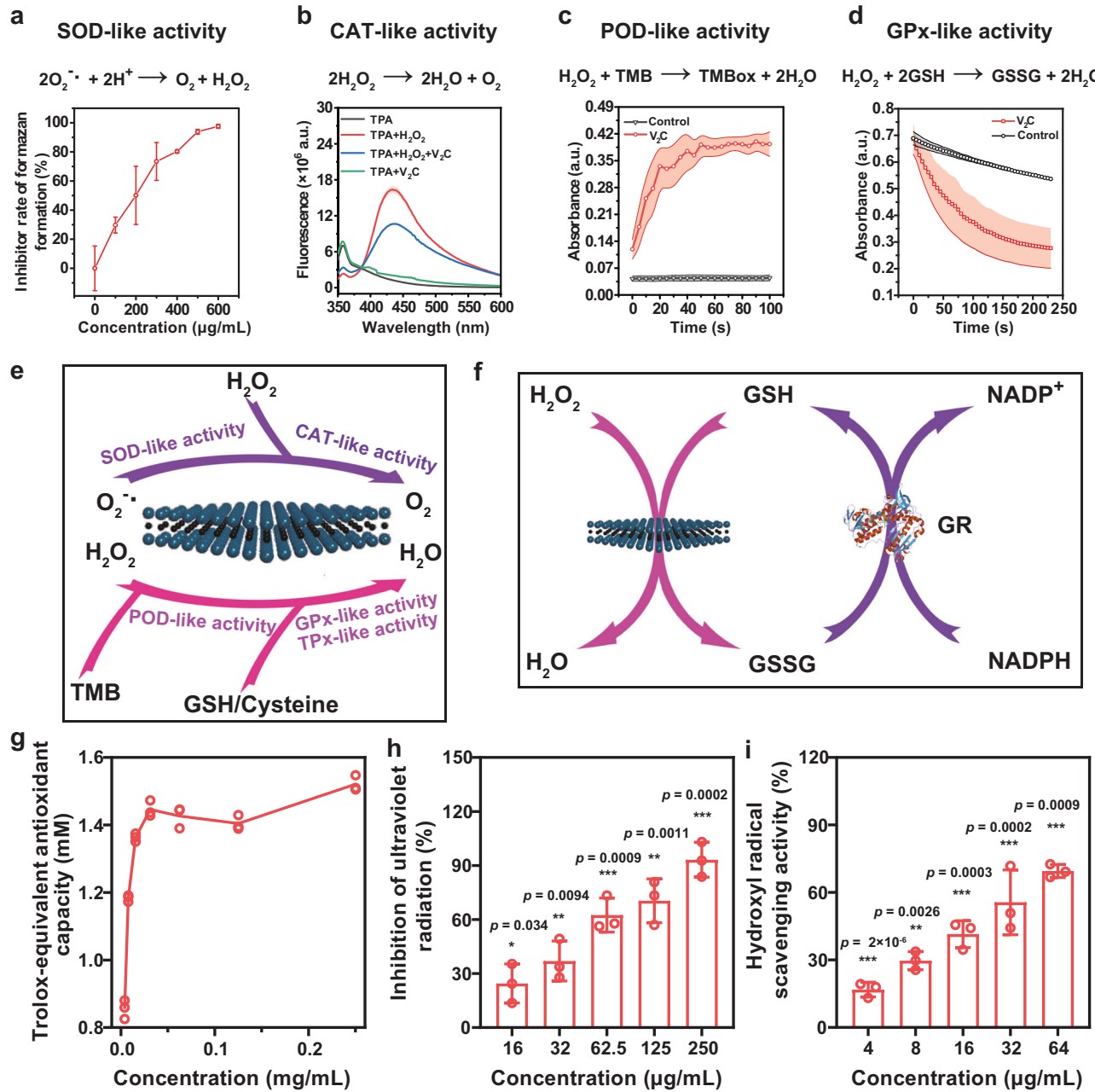

**Fig. 4 Multiple enzyme-mimicking activities of 2D V$_2$C MXenzyme. a** SOD-like activity of V$_2$C MXenzyme ($n = 3$ for each group). **b** CAT-like activity of V$_2$C MXenzyme ($n = 3$ for each group). **c** POD-like activity of V$_2$C MXenzyme ($n = 3$ for each group). **d** GPx-like activity of V$_2$C MXenzyme ($n = 3$ for each group). **e** Schematic illustration of enzyme-mimicking activities of V$_2$C MXenzyme. **f** Schematic representation exemplifying GPx-like activity of V$_2$C MXenzyme and GSH recycling by GR. **g** Total antioxidant capacity of V$_2$C MXenzyme ($n = 3$ for each group). **h** Ultraviolet radiation-inhibition activity of V$_2$C MXenzyme ($n = 3$ for each group). **i** Hydroxyl radical-scavenging activity of V$_2$C MXenzyme ($n = 3$ for each group). Data presented as mean ± SD, and asterisks indicate a significant difference (*$p < 0.05$, **$p < 0.01$, and ***$p < 0.001$) as compared with the control group using one-way analysis of variance (ANOVA).

Furthermore, assisted by other thiol-containing small molecules such as cysteine, V$_2$C MXenzyme is endowed with an apparent TPx-mimicking behavior by catalyzing the reduction of H$_2$O$_2$ to water (Supplementary Fig. 13), indicating that V$_2$C MXenzyme could utilize other thiol-molecules without GSH, which considers the factual situation that the GSH homeostasis is commonly disturbed under oxidative stress[35]. It is worth noting that, in the presence of halides (such as Br$^-$), V$_2$C MXenzyme also exhibits evident HPO-like activity by catalyzing the reduction of H$_2$O$_2$ to H$_2$O and the oxidation Br$^-$ to form the corresponding hypohalous acid (HOBr) (Supplementary Fig. 14). Taken together, these

results provide compelling evidence that the as-constructed 2D V$_2$C MXenzyme conducts six enzyme-mimicking activities under the physiological condition, including SOD, CAT, POD, GPx, TPx, and HPO (Fig. 4e), which is superior as compared to Cu$_x$O (three enzyme-like activities for CAT, GPx, and SOD)[6], Prussian blue (three enzyme-like activities for CAT, POD, and SOD)[33], Fe–N/C single-atom (four enzyme-like activities for POD, oxidase, CAT, and GPx)[36] and N-doped porous carbon nanospheres (four enzyme-like activities for oxidase, POD, CAT, and SOD)[11], demonstrating that 2D V$_2$C MXenzyme holds a broad-spectrum ROS removal capacity.

**The ROS-scavenging activity of 2D $V_2C$ MXenzyme**. It is essential to assess the ROS-scavenging activity of $V_2C$ MXenzyme and subsequently predict its potential for oxidation resistance. Total antioxidant capacity (TAC), an important parameter for assessing the antioxidant potentials, is usually expressed as Trolox (TR) and Vitamin C equivalents. The results reveal that TAC value escalates with the elevated concentrations of $V_2C$ MXenzyme (Fig. 4g, Supplementary Fig. 15). The ROS-scavenging activity of $V_2C$ MXenzyme was further assessed by ultraviolet (UV) protection activity since UV radiation has been used for inducing ROS generation such as •OH and $H_2O_2$[37]. Specifically, an apparent color fading is visually perceived for the reaction system treated by $V_2C$ MXenzyme (Supplementary Fig. 16a), and the corresponding absorbance of the product oxTMB in the reaction system with $V_2C$ MXenzyme is remarkably lower than that in the absence of $V_2C$ MXenzyme (Supplementary Fig. 16b), which reveals the UV protection capacity of $V_2C$ MXenzyme. This activity is concentration-dependent, where a higher concentration of $V_2C$ MXenzyme results in stronger UV protection efficacy (Fig. 4h). It is known that •OH is the most formidable kind of physiologically relevant ROS. Accordingly, the •OH-scavenging activity of $V_2C$ MXenzyme was also examined, which demonstrated that the signal intensity of •OH was notably decreased after the treatment with $V_2C$ MXenzyme (Supplementary Fig. 17a–c). Nearly 70% of •OH was eliminated after exposure to 64 µg mL$^{-1}$ $V_2C$ MXenzyme (Fig. 4i). Overall, the constructed 2D $V_2C$ MXenzyme possesses robust antioxidant enzyme-like activities and can efficiently scavenge miscellaneous ROS.

**2D $V_2C$ MXenzyme for in vitro protecting cells against oxidative stress**. After polyvinyl alcohol (PVA) modification on the surface, the 2D $V_2C$ MXenzyme displays high stability in various physiological solutions, including water, phosphate buffer solution (PBS), fetal bovine serum (FBS), and cell culture medium, with narrow size distribution and no obvious aggregation, guaranteeing further biomedical applications (Supplementary Figs. 18–20). Cell counting kit-8 (CCK-8) assay was performed to assess the cytocompatibility of $V_2C$ MXenzyme. Different cell types such as murine L929 fibroblasts and phaeochromocytoma cells (PC12) showed no visible cytotoxicity even the $V_2C$ MXenzyme was added at as high as 200 µg mL$^{-1}$ (Supplementary Fig. 21). The cellular uptake of both L929 and PC12 cells towards $V_2C$ MXenzyme was carried out to reveal their internalization behavior. After 24 h $V_2C$ MXenzyme exposure, the presence of V element in both cells was verified by EDX analysis (Supplementary Fig. 22a, b, e, f). TEM imaging also confirms the intracellular location of the $V_2C$ MXenzyme (Supplementary Fig. 22c, d, g, h). It is visible that $V_2C$ MXenzyme is mainly located at endocytic vesicles, suggesting internalization via the endocytic pathway. It is found that either amiloride or nystatin has a significant inhibitory effect on endocytosis of RB-labeled $V_2C$ MXenzyme, revealing that the cellular uptake of $V_2C$ MXenzyme in both L929 and PC 12 cells involves caveolae/lipid-mediated endocytosis and micropinocytosis (Supplementary Fig. 23). In addition, the amount of $V_2C$ MXenzyme in both cells was quantified by inductively coupled plasma optical emission spectroscopy (ICP-OES, Supplementary Fig. 24). Both cells display a time-dependent accumulation of V element levels when compared with the blank controls.

Subsequently, we provided definitive identification that $V_2C$ MXenzyme could act as an effective inhibitor of UV-modulated ROS production and thus reduce the damage and cytotoxicity of UV irradiation on both L929 cells and PC12 cells (Fig. 5a, b). $H_2O_2$ induced obvious concentration-dependent cytotoxicity on both L929 cells and PC12 cells since it could easily pass through the cytomembrane (Fig. 5c, e). Notably, the addition of $V_2C$

MXenzyme to the cell-culture medium effectively attenuates $H_2O_2$-mediated oxidative damage and maintains cell viability, which is also $V_2C$ MXenzyme concentration-dependent (Fig. 5d, f). Fenton-like reaction, which produces highly toxic •OH from the reaction between $H_2O_2$ and ferrous ion ($Fe^{2+}$), has been broadly employed to induce cancer cell death (Fig. 5g, h)[38]. Distantly, $V_2C$ MXenzyme induces an apparent decrease in the production of ROS as caused by Fenton reagent, as well as reduces Fenton regent-induced apoptosis, indicating the specific capability to eliminate ROS. Confocal laser scanning microscopy (CLSM) analysis further validates that $V_2C$ MXenzyme effectively mitigates oxidative stress and thus protects cells from ROS-induced cytotoxicity (Fig. 6).

To ascertain the cofactor influence, the intracellular GSH and GSSG levels were further monitored after different treatments (Fig. 5i, j). Buthionine sulfoximine (BSO), a compound that blocks GSH biosynthesis[25], reduces the cellular GSH levels. Although no distinct differences in the GSH and GSSG levels were found after treatment with $H_2O_2$ or $V_2C$ MXenzyme alone, the addition of both $H_2O_2$ and $V_2C$ MXenzyme with BSO could significantly diminish endogenous GSH level. Unlike the irreversible inhibition of BSO, the increased GSSG content did not lead to much decreased GSH level in the presence of $V_2C$ MXenzyme, which ascertains that $V_2C$ MXenzyme only employs GSH as a cofactor to get rid of $H_2O_2$, without disturbing the transformation from GSSG to GSH. As a result, the efficient intracellular uptake, noticeable ROS elimination, and impressive cytoprotection of $V_2C$ MXenzyme enable it as an effective artificial antioxidant enzyme to treat ROS-related diseases.

**Protection of ROS-induced damage of intracellular components**. The eventual outcome of ROS overproduction is oxidative stress-mediated apoptosis. As apoptotic effectors, a remarkable increment of caspase-3/7 activities was demonstrated to be involved in ROS-associated cytotoxicity in both L929 cells (Supplementary Fig. 25) and PC12 cells (Fig. 7a) by CLSM imaging. It is noted that the treatment with $V_2C$ MXenzyme results in a perceptible fluorescence reduction. Subsequently, according to the results of the Western blot assay, ROS-induced apoptosis as evidenced by the expression of cleaved caspase-3 (Fig. 7b). Furthermore, the levels of cleaved caspase-3 activity for Fenton's reagent treatment are obviously higher than $H_2O_2$ or UV treatment, further confirming the high toxicity of •OH radicals. However, all the expressions of cleaved caspase-3 activity are immensely decreased by the protection of $V_2C$ MXenzyme, which is ascribed to its ROS-scavenging capacity, especially for •OH.

Intracellular ROS overproduction induces the oxidative damage of three kinds of bioactive macromolecules including proteins, lipids, and DNA[39,40]. Subsequently, we explored the cytoprotective effect of $V_2C$ MXenzyme on intracellular protein carbonylation, lipid peroxidation, and DNA damage. Protein carbonylation, as the most common biomarker for ROS-induced irreversible protein damage, is usually formed on the side amino acid chains of lysine, proline, arginine, threonine, and tryptophan, which arouses conformational alternations in protein and directly leads to the loss of protein function[41]. The administration of $V_2C$ MXenzyme dramatically reduces the level of intracellular protein carbonylation, indicating $V_2C$ MXenzyme plays a vital role in defending protein against the accumulation of carbonylated groups induced by ROS (Fig. 7c, d). Lipid peroxidation, an irreversible process associated with cell or tissue necrosis, usually occurs late in the oxidative stress-induced injury[42]. The lipophilic fluorescent dye C11-BODIPY$^{581/591}$ was used as a ratiometric probe for the determination of lipid oxidation, which readily enters the cells and undergoes fluorescence emission shifts from

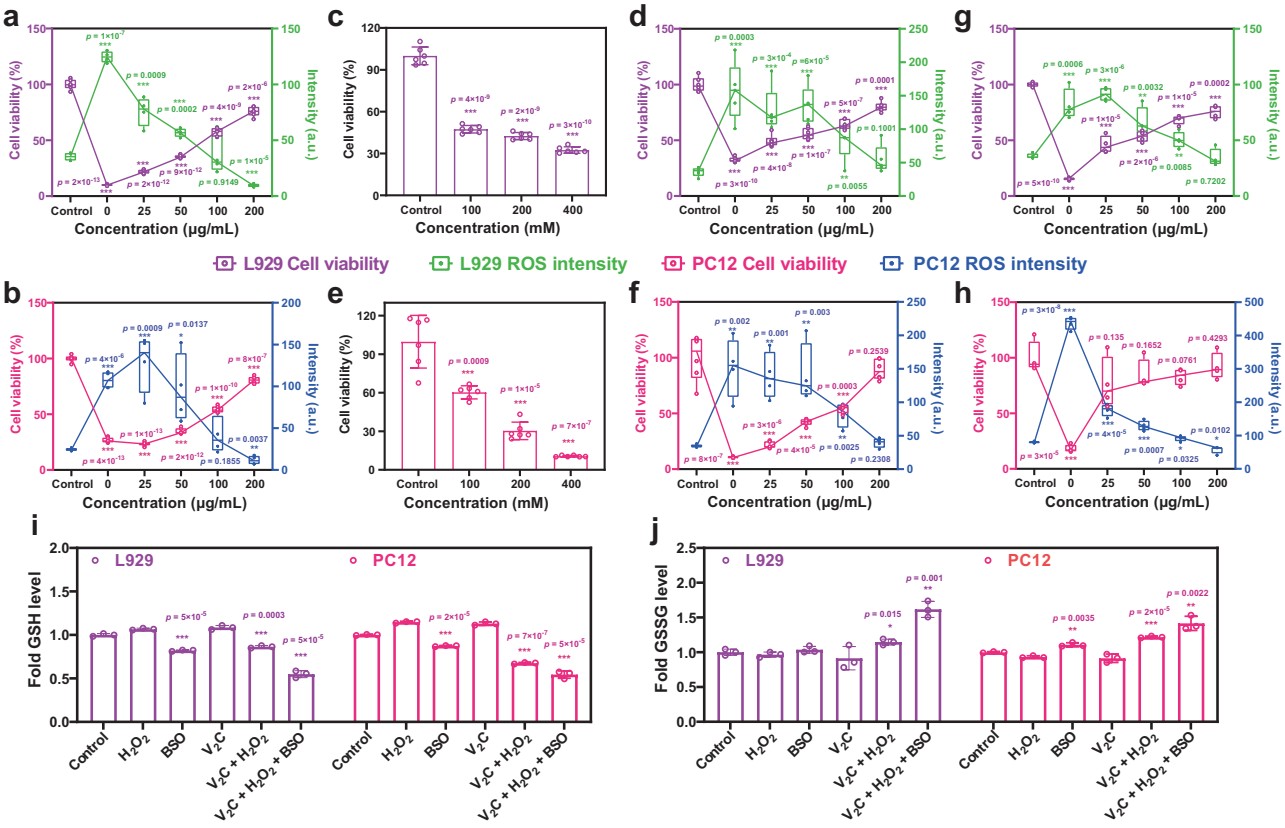

**Fig. 5 Effect of V₂C MXenzyme on intracellular ROS scavenging and cytoprotection.** $V_2C$ MXenzyme protects L929 cells from oxidative stress as induced by **a** UV irradiation ($n = 6$ for each group in cell viability and $n = 4$ for each group in ROS intensity), **c**, **d** $H_2O_2$ ($n = 5$ for each group in (**c**), $n = 6$ for each group in (**d**) cell viability and $n = 5$ for each group in (**d**) ROS intensity), and **g** Fenton reagent ($FeSO_4 + H_2O_2$, $n = 4$ for each group) via ROS scavenging. $V_2C$ MXenzyme protects PC12 cells from oxidative stress as induced by **b** UV irradiation ($n = 6$ for each cell viability group and $n = 4$ for each ROS intensity group), **e**, **f** $H_2O_2$ ($n = 6$ for each group in (**e**), $n = 6$ for each group in (**d**) cell viability and $n = 4$ for each group in (**d**) ROS intensity), and **h** Fenton reagent ($n = 4$ for each group). The central line represents the median (50th percentile), box limits represent the 25th and 75th percentiles and whiskers represent 1.5× the extent of the interquartile range. The cellular **i** GSH and **j** GSSG levels in both L929 cells and PC12 cells after different treatments ($n = 3$ for each group). Data presented as mean ± SD, and asterisks indicated significant difference (*$p < 0.05$, **$p < 0.01$, and ***$p < 0.001$) as compared with the control group using one-way analysis of variance (ANOVA).

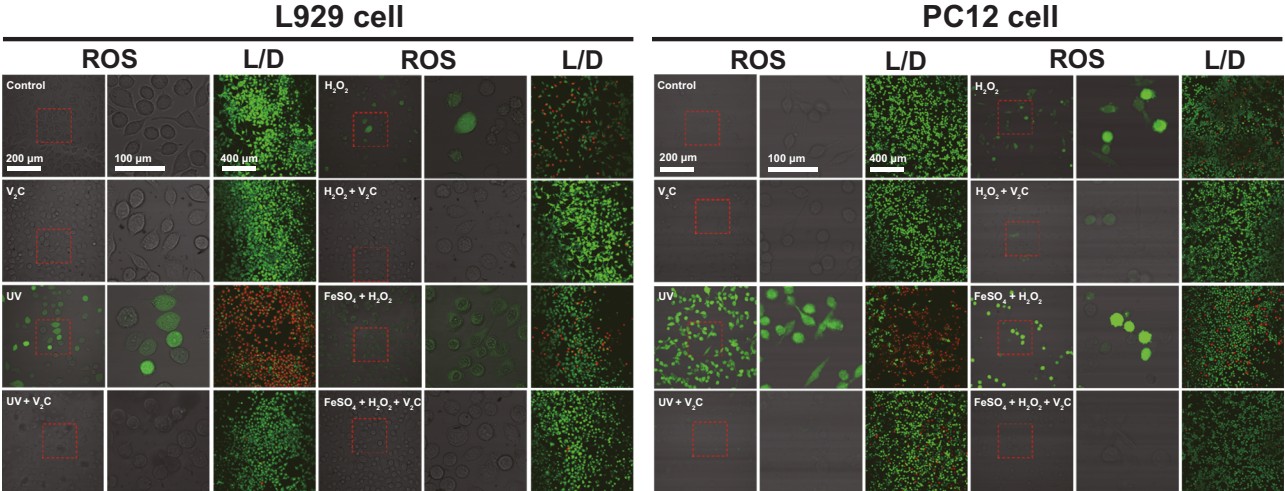

**Fig. 6 Effect of V₂C MXenzyme on intracellular ROS scavenging and cytoprotection.** CLSM images of L929 and PC12 cells with different treatments stained with DCFH-DA or Calcein-AM/PI (L/D represents Live/Dead). A representative image of three replicates from each group is shown.

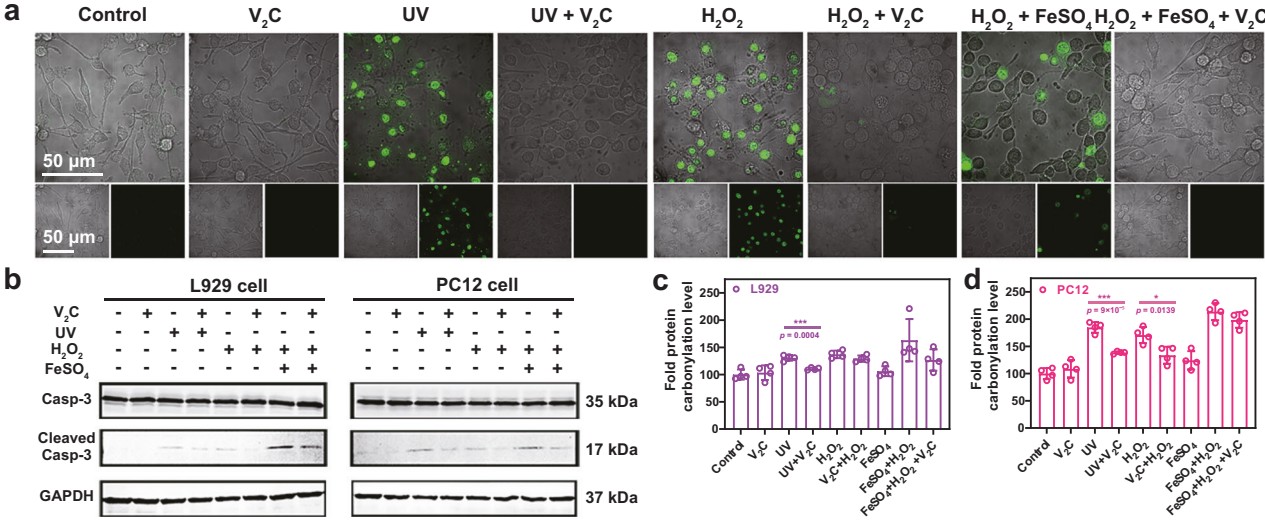

**Fig. 7 Protection of cellular components from ROS-induced damage by V₂C MXenzyme. a** CLSM images of Caspase-3/7 activity in PC12 cells after different treatments. A representative image of three replicates from each group is shown in (**a**). **b** Western blot analysis of caspase-3, cleaved caspase-3, and GAPDH expressions in both L929 cells and PC12 cells after different treatments. Protein carbonylation level of both **c** L929 cells and **d** PC12 cells after different treatments ($n = 4$ for each group). Data presented as mean ± SD and asterisks indicated significant difference (*$p < 0.05$, **$p < 0.01$, and ***$p < 0.001$) as compared with the control group using one-way analysis of variance (ANOVA).

red to green when attacked by the product of lipid peroxidation[43]. C11-BODIPY[581/591]-loaded cells with the stimulation of $H_2O_2$, UV irradiation, and Fenton's reagent demonstrate the anticipated red fluorescence decline and green fluorescence rise (Fig. 8a and Supplementary Fig. 26). Nevertheless, the V₂C MXenzyme-treated groups display no distinct difference in the red fluorescence, triumphantly indicating that V₂C MXenzyme can alleviate the intracellular oxidative stress and inhibit the formation of lipid peroxidation.

ROS can induce several types of DNA damage, among which DNA double-strand breaks (DSBs) can be potentially lethal to the cells[44]. Accordingly, we investigated the capability of the V₂C MXenzyme to protect DNA from ROS-induced DSBs. We analyzed the histone H2AX phosphorylated on Ser 139 (γ-H2AX) expression, which is a sensitive marker of DNA DSBs. Remarkably, an increment in γ-H2AX foci formation was monitored in the V₂C MXenzyme-untreated groups including $H_2O_2$, UV irradiation, and Fenton's reagent, whereas an apparent decrease was observed in the cells pretreated with V₂C MXenzyme, demonstrating that V₂C MXenzyme confers a protective effect on DNA integrity (Fig. 8b and Supplementary Fig. 27). Moreover, highlighting the ROS-scavenging capability of V₂C MXenzyme, cleavage of calf thymus DNA is substantially suppressed by pretreatment with V₂C MXenzyme in the presence of Fenton's reagent (Supplementary Fig. 28).

**In vivo anti-inflammation activity of V₂C MXenzyme.** The in vivo toxicity of 2D V₂C MXenzyme was initially evaluated. The results of established hematology, serum biochemistry, and histological assessment reveal that no appreciable inflammation, hydropic degeneration, pulmonary fibrosis, hyperplasia, necrosis, and other abnormal phenomena were examined in the groups for 4-weeks monitoring after intravenous injection of V₂C MXenzyme at the dose of 15 mg kg$^{-1}$ (Supplementary Figs. 29 and 30), collectively confirming that V₂C MXenzyme is safe within our tested dosage for further in vivo use.

Furthermore, acute local irritant dermatitis in the ear was initially established in mice by the topical application of phorbol 12-myristate 13-acetate (PMA) (Fig. 9a), which induced a protein kinase C-mediated pronounced inflammatory response[45,46].

2′,7′-dichlorofluorescin-diacetate (DCFH-DA) was applied through in situ administration for in vivo detection of ROS correlated with inflammation. After PMA administration for six hours, the PBS-treated ear unveiled strong fluorescence (Fig. 9b), indicating the endogenous ROS generation. Comparatively, when V₂C MXenzyme was injected into the site following PMA challenge, there was a remarkable fluorescence decrease (a 56.5% reduction) in the inflammatory ear (Fig. 9b, c), suggesting that V₂C MXenzyme possesses desirable ROS scavenging capability in vivo. Compared with hematoxylin and eosin (H&E)-stained images of the normal mouse ear, inflammatory cell infiltration was visibly detected in the PMA-treated ear (Fig. 9d), which evidenced PMA-induced activation of the inflammatory cascade. However, after treatment with V₂C MXenzyme, the symptom of cutaneous inflammation was alleviated, demonstrating the high efficacy of V₂C MXenzyme against inflammation.

To further enumerate and characterize the anti-inflammatory effect of V₂C MXenzyme, another acute ankle inflammation model was stimulated by lipopolysaccharide (LPS) (Fig. 9e). After 5 h of LPS challenge, more than 1.3-fold growth in the ankle fluorescence intensity of LPS-treated mice was observed compared with the untreated mouse (Fig. 9f, g), which was attributed to the enhanced ROS production by immune cells through the activation of toll-like receptor 4[47,48]. In contrast, the V₂C MXenzyme-treated ankle showed enormously reduced fluorescence (a 27.3% reduction) (Fig. 9g), suggesting that V₂C MXenzyme effectively suppresses LPS-induced inflammation. Pathological examination of ankle sections stained with H&E verified that more lymphocyte infiltrations were observed in the LPS-stimulated ankle whereas the treatment of V₂C MXenzyme remarkably attenuated the inflammation responses elicited by LPS (Fig. 9h). These findings are in accordance with the anti-inflammatory activity of V₂C MXenzyme in PMA-induced ear inflammation.

**In vivo neuroprotection in MPTP-induced mice PD model.** Oxidative stress is associated with certain neurodegenerative diseases, such as Parkinson's disease (PD)[49]. Therefore, it encourages us to explore the therapeutic effects of V₂C MXenzyme against oxidative stress-mediated neurotoxicity. A PD mouse model was

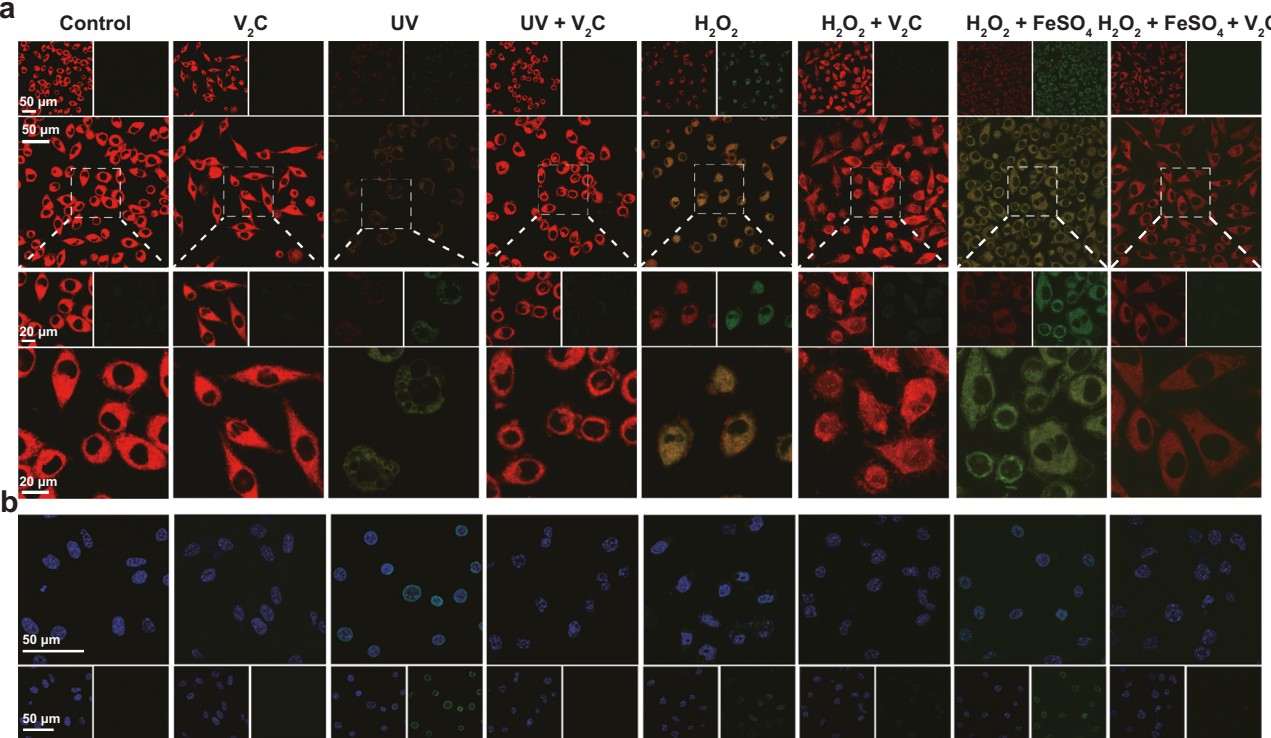

**Fig. 8 Protection of cellular components from ROS-induced damage by V₂C MXenzyme. a** CLSM image of lipid peroxidation of C11-BODIPY$^{581/591}$-stained L929 cells after different treatments. **b** Representative immunofluorescence CLSM images of γH2AX DNA damage foci in L929 cells after different treatments. A representative image of three replicates from each group is shown.

initially established by using 1-methyl-4-phenyl-1,2,3,6-tetra-hydropyridine (MPTP), which is the gold-standard agent for replicating almost all of the PD symptoms[50]. MPTP, as a lipophilic compound, can cross the blood–brain barrier, which was administered intraperitoneally. Thereafter, V₂C MXenzyme was implanted in solution into the striatum of MPTP-stimulated mice (Fig. 9i, Supplementary Fig. 31). Tyrosine hydroxylase (TH), the rate-limiting enzyme in dopamine biosynthesis, plays a vital role in the pathogenesis and treatment of PD[51]. Therefore, after different treatments, we further examined the alternations in the levels of striatal TH. Noteworthily, the TH levels were reduced remarkably in the MPTP-tread mice compared with those in the control group (Fig. 9j, k, Supplementary Figs. 32–34). In contrast, after V₂C MXenzyme treatment, the mice displayed considerably higher TH levels than only MPTP-stimulated mice, which indicated that V₂C MXenzyme helps to maintain TH activity and stability in parkinsonian mice. Compared with the untreated group, the upregulated expression of ionized calcium-binding adapter molecule 1 (IBA-1), a biological indicator of microglia activation, was distinctly observed in the striatum of the MPTP-treated group (Supplementary Figs. 35–37, Fig. 9l), reflecting that the increased release of pro-inflammatory cytokines from microglia could induce aggravated neuroinflammation[52]. On the contrary, V₂C MXenzyme treatment effectively inhibits IBA-1 expression, suggesting that V₂C MXenzyme could ameliorate the neuroinflammation of PD mice. Furthermore, it is commonly deemed that PD is ascribed to the striatal dopamine deficiency[51]. 4-Hydroxynonenal (4-HNE), a crucial bioactive marker of lipid peroxidation, is a protein adduct of oxidative stress[6]. We then testified V₂C MXenzyme for 4-HNE inhibition. As expected, compared to the MPTP-induced PD group, there is a remarkable reduction in the expression of 4-HNE in the V₂C MXenzyme-treated group (Supplementary Figs. 38–40, Fig. 9m), further indicating that V₂C MXenzyme treatment significantly protects mice from neurotoxicity by inhibiting MPTP-induced oxidative stress.

**Mechanism analysis of enzyme-mimicking activities.** The typical X-ray diffraction (XRD) pattern of V₂AlC exhibits the predominant peaks located at 13.47°, 35.55°, 36.24°, 41.27°, 45.28°, 55.52°, 63.86°, 75.27°, and 78.86°, which is indexed with JCPDS card No. 29-0101 (Fig. 10a). Compared with the MAX phase, a broad peak at around 6.05° corresponds to the (002) plane of MXene with a translation to the *c* lattice parameter of 14.5 Å, indicating the successful fabrication of V₂C MXene[53,54]. Because the surface chemistry is directly associated with the physico-chemical property, X-ray photoelectron spectroscopy (XPS), a well-established technique to assess the chemical composition and valence states of V₂C MXene, is performed to confirm the presence of V, C, O, and F elements (Supplementary Fig. 41a). Wherein, the V 2*p* region of V₂C MXene, ranging from 510 to 528 eV, is deconvoluted into four main peaks, which are assigned to V–C (513.2 eV), $V^{2+}$ (513.9 and 521.0 eV), $V^{3+}$ (515.8 eV), and $V^{4+}$ (523.0 eV) (Fig. 10b)[55,56]. The XPS C 1*s* spectrum is divided into five peaks of V–C, C–C, C–O, and O–C = O (Fig. 10c). The O 1*s* are fitted with four contributions at 529.3, 530.0, 531.3, and 532.5 eV, which are corresponding to V–$O_x$, V–O, C = O, and V–C–(OH), respectively (Fig. 10d)[55]. The F 1*s* peaks centered at binding energies of 684.3 and 686.1 eV are assigned to V–F and C–F bonds[55] (Supplementary Fig. 41b).

For the mechanism of V₂C MXenzyme as SOD mimetics, except C element, we infer that V may function as the catalytic component in the superoxide catalysis by the reaction between $V^{IV}$ and $V^V$ (Fig. 10e). Acting as an intermediate electron carrier, $V^{IV}$ can react with superoxide and generate vanadyl hydroperoxide (VOOH), where an electron transfer from $V^{IV}$ to $O_2^{-•}$. Then, the protonation of VOOH accompanies by the $H_2O_2$ release and $V^V$ regeneration.

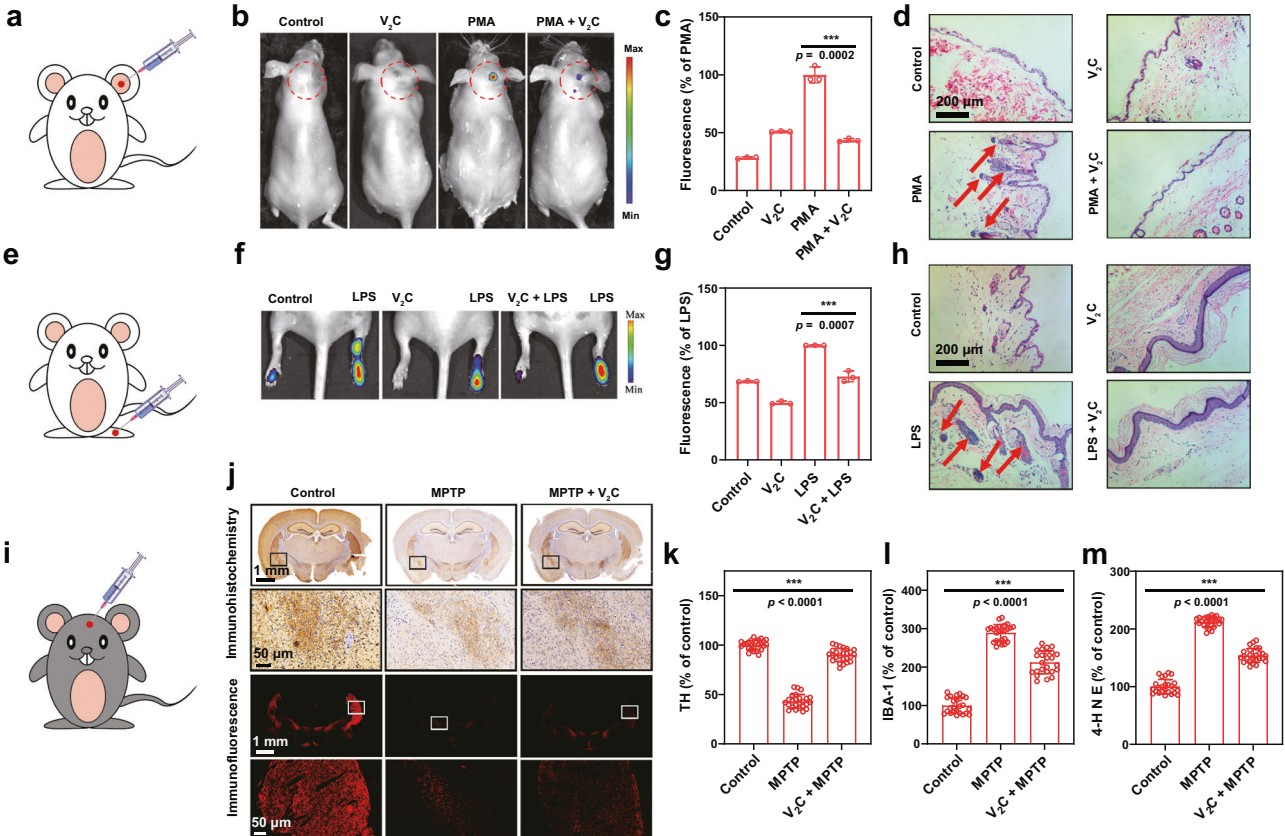

**Fig. 9 Inflammation and neurodegeneration therapy based on V₂C MXenzyme. a** Scheme of ear inflammation model. **b** In vivo fluorescence imaging of mice with different treatments to evaluate the effect of $V_2C$ MXenzyme on ROS scavenging in PMA-induced ear inflammation. **c** Corresponding radiant efficiency of the fluorescence images acquired in the live mice after different treatments ($n = 3$ for each group). **d** H&E-stained images of mice ears after different treatments. **e** Scheme of ankle inflammation model. **f** In vivo fluorescence imaging of mice with different treatments to evaluate the effect of $V_2C$ MXenzyme on ROS scavenging in LPS-induced ankle inflammation. **g** Corresponding radiant efficiency of fluorescence images acquired in the live mice after different treatments ($n = 3$ for each group). **h** H&E-stained images of mice ankles after different treatments. **i** Scheme of PD model treatment. **j** Immunohistochemistry and immunofluorescence images of TH expression in the brains of mice after different treatments (coronal plane). Expression levels of **k** TH, **l** IBA-1, and **m** 4-HNE in each treatment group (coronal plane) ($n = 25$ for each group), quantification represents the ratio of the experimental group to control. Data presented as mean ± SD and asterisks indicate significant differences (***$p < 0.001$) using one-way analysis of variance (ANOVA). A representative image of three replicates from each group is shown.

Thermodynamically, the dismutation of $O_2^{-\bullet}$ is concerned with the reduction and oxidation of $O_2^{-\bullet}$, where the redox potential values of $E(O_2/O_2^{-\bullet})$ and $E(O_2^{-\bullet}/H_2O_2)$ are 0.91 and −0.18 V, respectively[57]. As expected, the redox potential value of $V_2C$ MXenzyme reaches −0.11 V (Fig. 10f), which further confirms that the $V_2C$ MXenzyme is capable of catalyzing the $O_2^{-\bullet}$ dismutation.

Based on the fact that vanadium possesses a different redox state by complexing with $H_2O_2$, the CAT-like activity mechanism of the $V_2C$ MXenzyme is further revealed (Fig. 10g). When $H_2O_2$ is present in the reaction mixture, the $V^V$ species such as $OVO^+$ can be oxidized to form C1 monoperoxo vanadium species $OV(O_2)^+$[58]. Furthermore, the formed $OV(O_2)^+$ interaction with another $H_2O_2$ molecule results in the production of C2 diperoxo vanadium species $HOOV(O)_2^{2+}$. The reaction between $V^V$ and $H_2O_2$ for the generation of $V^{IV}$ and $\bullet OOH$ was evidenced in the electron spin resonance (ESR) spectrum measurement at liquid nitrogen temperature (−196 °C) (Supplementary Fig. 42). Followed by $OV^{2+}$ supplement, the variable C3 m-peroxo bridge $OVOOV(O_2)^+$ breaks up to generate $OVO^+$ and C4 oxo-peroxo radicals $\bullet OV(O_2)^{2+}$, which is ascribed to the internal oxidation. Finally, molecular oxygen ($O_2$) is released from the reaction system through the dismutation reaction of $\bullet OV(O_2)^{2+}$.

As evidenced in the Fourier transform infrared (FTIR) spectroscopy, the characteristic vibration of $V_2C$ MXenzyme at

1000 cm$^{-1}$ corresponds to the bond of V-oxo (V = O) (Supplementary Fig. 43)[25]. However, the vibration peak of V = O disappears after adding $H_2O_2$ into the reaction system, indicating that V-peroxo species might be formed. Meanwhile, in the Raman spectra, the emerging of a weak band at 1200 cm$^{-1}$ corresponds to V-peroxido species resulting from $V_2C$ MXenzyme treated with $H_2O_2$[25], verifying that the V peroxide species 1 is generated on the surface of $V_2C$ MXenzyme. After further GSH treatment, the band of V = O at 1000 cm$^{-1}$ in the FTIR spectrum and 1200 cm$^{-1}$ in the Raman spectrum does not reappear (Supplementary Fig. 44), suggesting that the V = O reproduction for $V_2C$ MXenzyme as GPx mimetics is not required, which proceeds along with another pathway. According to the proposed mechanism (Fig. 10h), $V_2C$ MXenzyme provides reaction sites for $H_2O_2$ reduction accompanied by GSH oxidation. Subsequently, GSH acting as a proton carrier protonates the partially negatively charged oxygen ($\delta-$), and $GS^-$ as the nucleophile attacks the positively charged oxygen ($\delta+$) in the meantime, generating a labile sulfonate-bound intermediate G2, which disintegrates a G3 glutathione sulfenic acid (GSOH) and a G4 dihydroxo intermediate through the hydrolysis. It is noted that the GSOH produced by this pathway is likely the migration of HOBr from an intermediate of V−OBr in vanadium-dependent haloperoxidase[24,25] (Supplementary Fig. 45). In the presence of

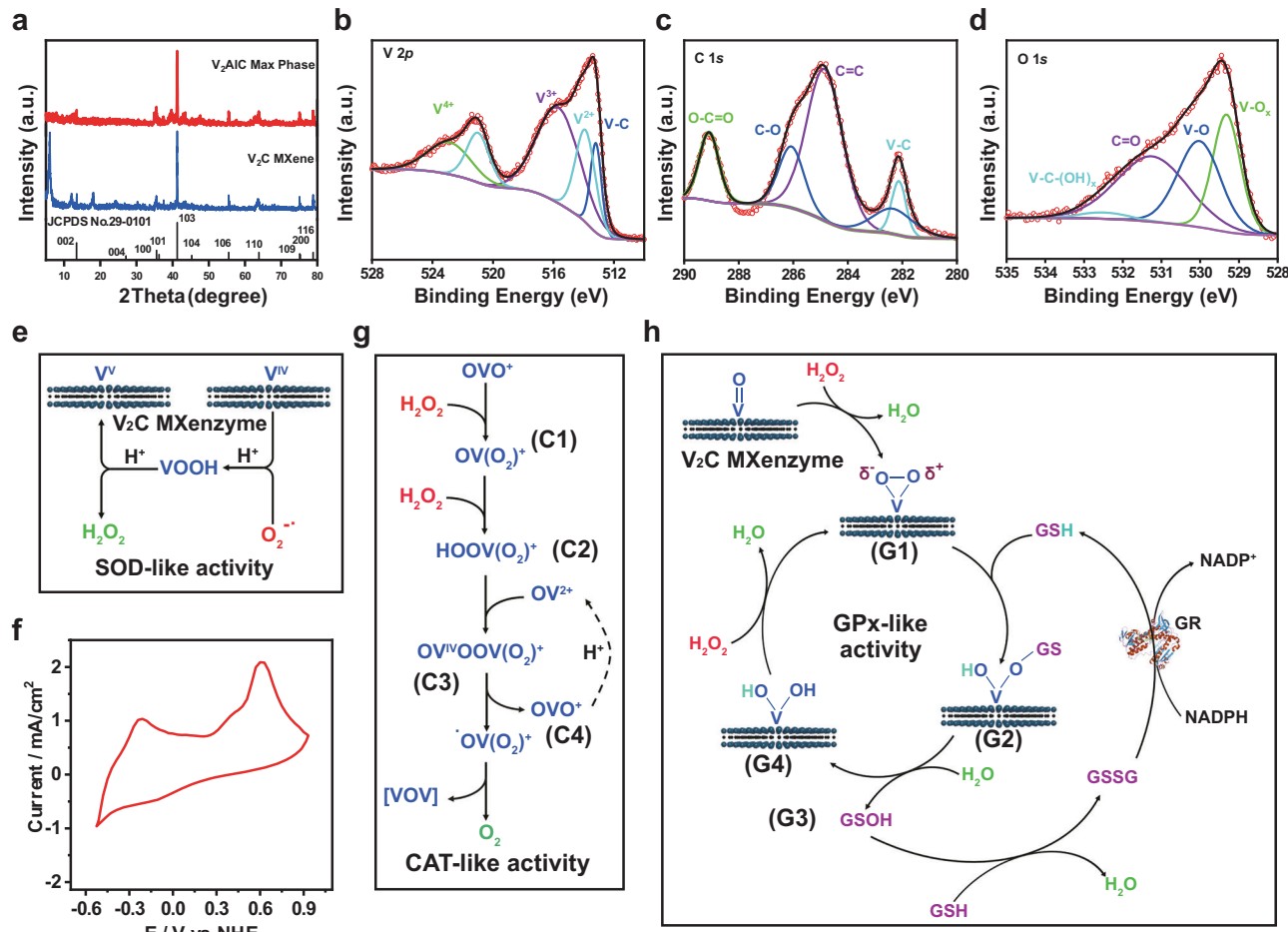

**Fig. 10 Mechanism investigation of enzyme-mimicking activities. a** XRD diffraction patterns of $V_2AlC$ MAX phase ceramic and $V_2C$ MXene. High-resolution XPS spectra of **b** V 2*p* region, **c** C 1*s* region, and **d** O 1*s* region. **e** Schematic illustration on clarifying the underlying mechanism of SOD-like activity of $V_2C$ MXenzyme. **f** Cyclic voltammogram of $V_2C$ MXenzyme showing their redox potential. **g** Schematic illustration revealing the related mechanism of CAT-like activity of $V_2C$ MXenzyme. **h** Schematic illustration unveiling the mechanism of GPx-like activity of $V_2C$ MXenzyme.

enough GSH, the GSOH reacts with GSH to generate GSSG, which can be reduced back to GSH by the GR/NADPH system. Besides GSH, other small molecules containing thiol groups (–SH), including cysteine, can be employed as thiol cofactors, which probably accounts for the TPx-like activity of $V_2C$ MXenzyme.

Finally, for the underlying mechanism of $V_2C$ MXenzyme as POD mimetics, in $V_2C$ MXene structural model, the V atom is supposed to act as Lewis acid site, but the bond pair of electrons for bridging oxygen atoms behaves as Lewis base sites, in which the nucleophilic addition reactions of oxygen happen. Subsequently, the $V_2C$ MXene is supposed to react with $H_2O_2$ to form an intermediate V-peroxo (P1) species (Supplementary Fig. 46), and then the TMB substrate binds to the V-peroxo complexes via nucleophilic attack to form P2, thus allowing the oxidation reaction of TMB to form the $TMB^{*+}$ species. Because $H_2O_2$ is a two-electron oxidant, another TMB molecule is required for $V_2C$ MXenzyme regeneration inducing TMBox formation.

## Discussion

In this work, we demonstrated the specific capability of 2D $V_2C$ MXenzyme to serve as robust multifunctional inorganic analogs of SOD, CAT, POD, TPx, GPx, and HPO, mimicking intracellular antioxidant defense system against ROS-mediated critical oxidative damage (e.g., protein carbonylation, lipid peroxidation, and DNA damages), which extends their biomedical use from traditional chemical catalysis and energy storage to neoteric catalytic biomedicine. Both in vitro and in vivo experiments verified that $V_2C$ MXenzyme not only possessed desirable biocompatibility but also exhibited impressive ROS-scavenging capability to protect cell components against oxidative stress through catalytic reactions. Taken together, our MXenzyme is acknowledged as a valuable toolkit for the specific utilization in multifarious inflammation and neurodegeneration treatment.

## Methods

**Materials and reagents**. Layered ternary vanadium aluminum carbide ($V_2AlC$, 200 mesh powders with 98% metals basis) was purchased from Forsman Scientific Co., Inc. (Beijing, China). TPAOH (40%), 5,5-dimethyl-1-pyrroline-N-oxide (DMPO), buthionine sulfoximine (BSO), trichloroacetic acid (TCA), guanidine hydrochloride, and TMB were obtained from Adamas-beta Inc. (Shanghai, China). Hydrogen peroxide ($H_2O_2$), hydrochloric acid (HCl), and HF were purchased from Sinopharm Chemical Reagent Co., Ltd. (Shanghai, China). Cysteine, 2-monochlorodimedone (MCD), 5,5′-dithiobis-(2-nitrobenzoic acid) (DTNB) and 1-methyl-4-phenyl-1,2,3,6-tetrahydropyridine (MPTP) were obtained from Aladin Ltd. (Shanghai, China). Polyvinyl alcohol (PVA, 87-90% hydrolyzed), bovine serum albumin (BSA), Triton X-100, phorbol 12-myristate 13-acetate (PMA), LPS, Rhodamine B (RB), amiloride, chlorpromazine, nystatin, and hematoxylin and eosin (H&E) were purchased from Sigma-Aldrich (Shanghai) Trading Co., Ltd. (Shanghai, China). Roswell Park Memorial Institute (RPMI) 1640 medium were obtained from Hyclone Laboratories (Logan, Utah, USA). Calcein AM and propidium iodide (PI) were both purchased from Shanghai Hongmao Biotechnology Co., Ltd. (Shanghai, China). ROS assay kit, glutathione peroxidase (GPx) assay kit, total antioxidant capacity (TAC) assay kit, cell counting kit-8 (CCK-8) assay kit, glutathione (GSH)/oxidized glutathione (GSSG) assay kit, 2-(4-Amidinophenyl)-6-indolecarbamidine dihydrochloride (DAPI) and radio-immunoprecipitation assay

(RIPA) buffer were obtained from Beyotime Institute of Biotechnology (Haimen, Jiangsu, China). Paraformaldehyde was purchased from Beijing Dingguo Changsheng Biotechnology Co., Ltd. (Beijing, China). C11-BODIPY (581/591) dye and CellEvent caspase-3/7 green detection reagent were obtained from Thermo Fisher Scientific Inc. (Waltham, MA, USA). Caspase-3 antibody (CST #9662), cleaved caspase-3 (CST #9661), and glyceraldehyde-3-phosphate dehydrogenase (GAPDH, CST #2118) antibody were purchased from Cell Signaling Technology Inc. (Danvers, MA, USA). Anti-tyrosine hydroxylase (Abcam #ab112), anti-4 hydroxynonenal (Abcam #ab46545), anti-IBA 1 (Abcam #178847) antibody were obtained from Abcam Inc. (Cambridge, MA, USA). Deionized (DI) water with a resistivity of 18.2 Ω was used throughout the study. All other reagents and chemicals were used as received from Adamas-beta Inc. (Shanghai, China) without further purification.

**Synthesis of V$_2$C MXenzyme.** To synthesize single- and few-layer V$_2$C MXenzyme, 1 g of MAX phase precursor V$_2$AlC powder was slowly immersed in 30 mL of 50 wt% concentrated aqueous HF solution at room temperature while stirring with a Teflon-coated magnetic bar for 72 h. After HF etching by removing the Al layers, the suspension was centrifuged at $2,292 \times g$ for 10 min and washed several cycles using DI water until the pH of the supernatant reached higher than 5.0. The bulked settled precipitation was obtained and stir-mixed with 40 mL of TPAOH aqueous solution for 24 h at room temperature. Eventually, the intercalated V$_2$C was purified by centrifugation and rinsed with argon (Ar) deaerated DI water three times to remove the residual TPAOH, followed by centrifuging at $1,467 \times g$ for 50 min and collecting the colloidal supernatant for further investigation.

**Surface modification with PVA.** The obtained clay-like V$_2$C was highly water-soluble but not stable in physiological solutions without further modification. For surface modification, 5 mL (2 mg mL$^{-1}$) of the as-synthesized V$_2$C solution was dropwise added into 10 mL of PVA aqueous solution (10 wt%). After ultrasonication for 20 min and vigorous stirring for 2 h, excess PVA in the V$_2$C-PVA sample was removed by centrifugation at $33,095 \times g$ and rinsed repetitively with DI water. Thereafter, the resulting PVA-modified V$_2$C was homogeneously re-dispersed in deaerated DI water and stocked under 4 °C before future use.

**Characterization.** TEM images and SAED were obtained using a JEM-2100F field emission transmission electron microscope at an accelerating voltage of 200 kV (JEOL Company Ltd., Japan). High-resolution STEM, corresponding EDS analysis, HAADF-STEM images and element mapping were conducted on a JEM-ARM 300F Grand ARM (JEOL Company Ltd., Japan) operated at 80 kV and equipped with two spherical aberration correctors. FESEM images, EDS, corresponding elemental mapping, and linear scanning were performed using a Magellan 400 field emission scanning electron microscope (FEI Company, USA). The amount of vanadium (V) element was tested with an Agilent 725 ICP-OES (Agilent Technologies, USA). XPS measurements were carried out on the ESCAlab250 electron spectrometers (Thermal Fisher VG, USA). XRD pattern was performed by using a Rigaku D/MAX-2200 PC XRD instrument (Rigaku Co. Ltd., Tokyo, Japan) equipped with Cu Kα radiation ($\lambda = 1.54$ Å). •OH generation was detected using DMPO as the spin trapping agent at room temperature on an A200S-95/12 ESR spectrometer (Bruker, Germany). FTIR spectra were collected with a Bruker Tensor II (Bruker Optics, Germany) FTIR analyzer. Raman spectra were determined by using a Renishaw in a microscopic confocal Raman microscope system (Renishaw, UK). AFM topography images were acquired by using a NTEGRA Prima (NT-MDT, Russia) AFM with Nova software under ambient conditions. The fluorescent images were captured by using a FluoView FV1000 confocal laser scanning fluorescence microscope (Olympus Company, Japan). Ultraviolet-visible-near-infrared (UV–vis–NIR) absorption spectra were recorded using a Shimadzu UV-3600 UV–vis–NIR scanning spectrometer (Shimadzu Scientific Instruments, Japan). The photoluminescence spectra were recorded using an FLS 980 spectro-fluorimeter (Edinburgh Instruments Ltd., England).

**SOD-like activity of V$_2$C MXenzyme.** The ability of V$_2$C MXenzyme quenching superoxide anion radical (O$_2^{-•}$) was investigated by measuring the inhibition of formazan formation through WST-1 using a colorimetric SOD assay kit according to the manufacturer's instructions. In brief, O$_2^{-•}$ was produced via the oxidation of xanthine with XOD. The amount of formazan generated is directly correlated to the number of O$_2^{-•}$ produced in the reaction system. Therefore, to verify the O$_2^{-•}$-scavenging ability of V$_2$C MXenzyme, WST-1, xanthine and XOD were mixed in V$_2$C dispersion with different concentrations (0, 25, 50, 100, 200, and 400 μg mL$^{-1}$), then the absorption changes of water-soluble formazan were spectrophotometrically monitored at 450 nm using UV–vis–NIR spectroscopy ($n = 3$ for each group).

**CAT-like activity of V$_2$C MXenzyme.** The CAT-like activity of the V$_2$C MXenzyme was evaluated by measuring inhibition of the yielding of 2-hydroxyterephthalic acid (TAOH) through fluorescence analysis. H$_2$O$_2$ (10 mM) can decompose into •OH, which are further captured by TPA (500 μM) to generate highly fluorescent TAOH with an emission peak around 425 nm upon excitation at the wavelength of 320 nm. In the presence of CAT or CAT mimics (250 μg mL$^{-1}$), H$_2$O$_2$ was catalytically decomposed into H$_2$O and O$_2$, which could not react with TA to produce

TAOH. Hence, the elimination of H$_2$O$_2$ was assessed by monitoring the fluorescence signal of TAOH ($n = 3$ for each group). The concentration of H$_2$O$_2$ (10 mM) consumption with V$_2$C (20 μg mL$^{-1}$) was determined by directly monitoring the absorption spectra at 240 nm using a UV-vis–NIR spectrophotometer ($n = 3$ for each group). The CAT-like activity was further evaluated based on the O$_2$ generation in the presence of H$_2$O$_2$ (10 mM). The produced O$_2$ (unit: mg L$^{-1}$) from the addition of H$_2$O$_2$ (2, 4, 10, 20, 40, 200, and 400 mM) was quantified by using a portable dissolved oxygen meter (INESA Scientific Instrument Co., Ltd., China).

**GPx-like activity of V$_2$C MXenzyme.** The GPx-like activity of the V$_2$C MXenzyme was studied spectrophotometrically using a GPx assay kit as per the manufacturer's instructions, which was based on a GR coupled assay. Briefly, GSH (8.4 mM) was oxidized to GSSG in the presence of H$_2$O$_2$ (5 mM) and V$_2$C (200 μg mL$^{-1}$), and then the reduction of GSSG to GSH was mediated by GR with reduced NADPH providing reducing power. Moreover, the amount of NADPH (300 μM) decreased, which was concomitant with a change in absorbance at 340 nm and quantified on UV–vis–NIR spectroscopy ($n = 3$ for each group).

**POD-like of V$_2$C MXenzyme.** The POD-like activity assay of the prepared V$_2$C MXenzyme was performed using TMB as the POD substrate with the assistance of H$_2$O$_2$. In a typical test, TMB (0.8 mM), H$_2$O$_2$ (1 mM), and V$_2$C (100 μg mL$^{-1}$) were mixed via pipetting at room temperature in an acetate buffer solution. As the reaction proceeded, the blue color was developed and the absorbance changes of the reaction mixture at 652 nm were immediately measured in time-scanning mode to assess the POD-like activity ($n = 3$ for each group).

**TPx-like activity of V$_2$C MXenzyme.** The TPx-like activity test of the synthesized V$_2$C MXenzyme was conducted using cysteine (2 mM), V$_2$C MXene (0, 50, 100, 200, and 400 μg mL$^{-1}$), and H$_2$O$_2$ (0, 0.8, 1.6, 3.2, 6.4, and 9.6 mM) in PBS. After every 0.5 min interval, 100 μL of the prepared mixture was placed into 100 μL DTNB (5 mM) in methanol solution, which was reacted for another 2 min and then diluted to 1 mL using PBS. As the reaction proceeds, the yellow color reduced and the absorbance of DTNB was measured at 412 nm. The control reactions were carried out in the absence of V$_2$C MXenzyme ($n = 3$ for each group).

**HPO-like activity of V$_2$C MXenzyme.** The HPO-like activity of V$_2$C MXenzyme was carried out in the presence of H$_2$O$_2$ (10 μM), Br$^-$ (1 mM), V$_2$C MXene (50 μg mL$^{-1}$), and MCD (50 μM). The absorbance changes of the reaction mixture at 290 nm were immediately measured in time-scanning mode to assess the HPO-like activity. The control reactions were carried out in the absence of V$_2$C MXene ($n = 3$ for each group).

**Steady-state kinetic analysis.** The steady-state kinetic studies were carried out by recording the absorbance of the reaction systems in time-course mode at room temperature using a UV–vis–NIR spectrophotometer. The apparent kinetic parameters were calculated by nonlinear least-squares fitting the absorbance data based on the Michaelis-Menten equation, which indicates the relationship between the rates of substrate conversion by an enzyme and the concentration of the substrate, as follows: $v = V_{max}[S]/(K_m + [S])$, where $v$ is the initial reaction velocity, $V_{max}$ stands for the maximum conversion velocity, [S] represents the concentration of the substrate, and $K_m$ corresponds to the Michaelis constant. $K_m$ is equivalent to the substrate concentration once the rate of conversion reaches half of $V_{max}$, which reveals the affinity of the enzyme to its substrate: the smaller $K_m$ value and the higher affinity of the enzyme.

**TAC assay.** The direct antioxidant capacity of V$_2$C MXenzyme was evaluated by using a TAC assay kit based on a rapid 2,2-Azino-bis-3-ethylbenzothiazoline-6-sulfonic acid (ABTS) radical cation (ABTS•$^+$) decolorization method according to the manufacture's instruction. Briefly, ABTS stock solution was mixed with potassium persulfate and stored in dark at room temperature for 12 h to generate ABTS•$^+$. The resulting dark blue-green radical mixture was diluted to adjust the absorbance to around 0.7 at 734 nm before use. Trolox (0.15, 0.3, 0.6, 0.9, and 1.2 mM), a water-soluble analog of vitamin E and vitamin C (0.15, 0.3, 0.6, 0.9, 1.2, and 1.5 mM), were employed as a reference standard for a calibration curve. Thereafter, the above diluted ABTS•$^+$ working reagent was mixed with various volumes of Trolox, vitamin C, or sample for 10 min. The detection absorbance was at 734 nm. Finally, the TAC results of V$_2$C (3.9, 7.8, 15.6, 31.3, 62.5, 125, and 250 μg mL$^{-1}$) were calculated by using the standard curve and expressed as mM Trolox-equivalent antioxidant capacity ($n = 3$ for each group).

**Ultraviolet radiation inhibition activity.** Totally, 0.8 mM of TMB solutions were placed in 1.5 mL Eppendorf tubes containing V$_2$C MXenzyme with different concentrations (0, 16, 32, 62.5, 125, and 250 μg mL$^{-1}$), and irradiation was carried out under a 365 nm ultraviolet lamp (Lichen, China) in a distance of 10 cm at a dose of 20 J cm$^{-2}$ for 30 min. After that, the absorption changes of the reaction mixture were spectrophotometrically monitored at 652 nm using UV–vis–NIR spectroscopy ($n = 3$ for each group).

**•OH scavenging activity**. The synthesized $V_2C$ MXenzyme was employed for scavenging •OH by using ESR spectroscopy at room temperature. Furthermore, the •OH generation was initiated by utilizing Fenton-like reagents, and trapped as spin adduct DMPO/•OH. In brief, ESR measurement was assessed by incubating $FeSO_4$ with $H_2O_2$ aqueous solution followed by adding $V_2C$ and DMPO. Subsequently, the ESR spectra were recorded after 2 min of incubation.

•OH-scavenging capacity of the samples was also assessed via the salicylic acid (SA) method using a UV–vis–NIR spectrophotometer. Typically, •OH was typically introduced through the Fenton reaction of 2 mM $FeSO_4$ and 5 mM $H_2O_2$ for 5 min followed by adding $V_2C$ MXenzyme into the solution. Finally, the remained amount of •OH was detected by examining the characteristic absorption peak of 2,3-dihydroxybenzonic acid at 510 nm, which was formed from the oxidation of SA with •OH ($n = 3$ for each group).

**Cell culture**. Two different cell models were used in vitro. The mice fibroblast cell line (L929 cells), which originates from an immortalized mouse fibroblast cell line, is the internationally recognized cell that is routinely used in in vitro cytotoxicity assessments. A neuronal cell line (PC12 cells), derives from the neural crest that has a mixture of neuroblastic cells and eosinophilic cells and can differentiate into neuron-like cells by nerve growth factor stimulation. PC12 cells have been widely used as a cellular model for neuronal development and neurological diseases because of their neuronal properties. Both the L929 mouse fibroblasts cell line and PC12 pheochromocytoma cell line were originally obtained from the Cell Bank of Shanghai Institutes for Biological Sciences, Chinese Academy of Sciences (Shanghai, China). All cells were regularly maintained in complete RPMI-1640 medium supplemented 10% FBS, 100 U mL$^{-1}$ penicillin, and 100 U mL$^{-1}$ streptomycin. Both cultures were kept at 37 °C in a humidified atmosphere of 5% $CO_2$ and 95% air. The medium was changed every two days, and the cells were routinely harvested by using 0.25% trypsin solution before approached 80% confluence.

**Cell viability study**. The cytotoxicity of $V_2C$ MXenzyme was evaluated by CCK-8 assay. Briefly, 100 µL of L929 and PC12 cells in growth medium were placed in each well of 96-well plates at a density of $8 \times 10^4$ cells mL$^{-1}$ at 37 °C. After incubation for 24 h, the old culture medium was replaced with 100 µL fresh one containing $V_2C$ MXenzyme at various concentrations (0, 12.5 25, 50, 100, 200, and 400 µg mL$^{-1}$). After another 24 h culture, the cells were rinsed twice lightly with prewarm PBS. Totally, 100 µL of fresh culture media containing 10 µL of CCK-8 work solution was then added and treated for further 2 h. Subsequently, the relative cell viability was calculated by measuring the absorbance of CCK-8 at 450 nm using a Spectra M2 microplate reader (Molecular Devices, CA, USA) ($n = 6$ for each group).

**Intracellular ROS detection**. The intracellular ROS level was determined using a ROS assay kit based on the oxidant-sensitive fluorescent dye DCFH-DA as per the manufacturer's instructions. DCFH-DA as a non-fluorescent cell-permeable compound diffuses across cell membranes and is hydrolyzed by intracellular esterases to form 2′,7′-dichlorodihydrofluorescin (DCFH). After that, two-electron oxidation of DCFH by ROS results in the generation of a highly fluorescent product 2′,7′-dichlorofluorescein (DCF). To conduct the experiment, pretreated cells were incubated with a 1 mL serum-free culture medium containing 10 µM of DCFH-DA. After incubation for 30 min at 37 °C, the cells were washed with warm PBS three times to remove the excess dye. Finally, the fluorescence intensity of each well was measured using a Spectra M2 microplate reader, and fluorescence visualization was carried out by a CLSM. For DCF detection, the excitation wavelength was at 488 nm, and the emission peak was examined at 525 nm.

**Protecting cells from Fenton's reagent-induced oxidative stress**. Cells were first inoculated in 96-well plates ($1 \times 10^4$ cells per well) or 20 mm glass-bottom cell culture dishes ($1 \times 10^5$ cells per dish). After incubation for 24 h, the old medium was discarded and the cells were exposed to fresh medium containing $H_2O_2$ (0, 100, 200, and 400 mM) or Fenton's reagent (400 µM of $H_2O_2$ and 40 µM of $FeSO_4$) in the presence of different concentrations of $V_2C$ MXenzyme (0, 25, 50, 100, and 200 µg mL$^{-1}$) for another 12 h incubation. In order to induce the increased oxidative stress, $H_2O_2$ or Fenton's reagent should be freshly prepared from a stock solution before each experiment. Finally, the cell viability and ROS level were detected using CCK-8 and ROS assay, respectively ($n = 4$ for each group).

**Protecting cells from UV-induced oxidative stress**. Cells were added into 96-well plates ($1 \times 10^4$ cells per well) or 20 mm glass-bottom cell culture dishes ($1 \times 10^5$ cells per dish), and cultured for 24 h. Subsequently, the medium was replaced with a fresh growth medium containing series of concentrations from 0 to 200 µg mL$^{-1}$ of $V_2C$ MXenzyme for another 12 h. After that, the medium was replaced with a fresh growth medium and the cells were irradiated under a 365-nm UV lamp (Lichen, China) at a dose of 20 J cm$^{-2}$ at a distance of 10 cm. After irradiation for 30 min, the cells were incubated for 4 h, and then the cell viability ($n = 6$ for each group) and ROS level ($n = 4$ for each group) were measured, respectively.

**Assessment of intracellular GSH and GSSG levels**. To assess the intracellular levels of GSH and GSSG, cells were seeded into 6-well plates at the density of $2 \times 10^5$ cells at 37 °C. After treatment with $V_2C$ (200 µg mL$^{-1}$), $H_2O_2$ (200 µM), and BSO (50 µM), respectively, cells were collected, rinsed with PBS and freeze–thaw breakage with liquid nitrogen and 37 °C water bath, followed by total GSH/GSSG quantification using GSH/GSSG assay kit according to the manufacturer's instructions ($n = 3$ for each group).

**Internalization of $V_2C$ MXenzyme in mammalian cells**. To measure the internalization of $V_2C$ MXenzyme, cells were placed into 6-well plates at the density of $2 \times 10^5$ cells and routinely cultured at 37 °C for 24 h. Subsequently, the medium was removed and the cells were treated with $V_2C$ MXenzyme (600 µg mL$^{-1}$) for 6 h, and the cells were washed with PBS and fixed with 2.5% glutaraldehyde solution (1 mL) at 4 °C for 20 min followed by dehydration through in serial gradient alcohol. Furthermore, the cells were lyophilized and morphology images were acquired by SEM. The vanadium content in the sample was detected using EDX spectroscopy.

To quantitate intracellular uptake of $V_2C$ MXenzyme, the cells were incubated with $V_2C$ MXenzyme (600 µg mL$^{-1}$) for 6 h, followed by cell lysis and ionization in the presence of aqua regia for 2 h. Finally, the cell samples were quantified by using ICP-OES ($n = 3$ for each group).

$V_2C$ MXenzyme was labeled by a red fluorescence dye RB by the following methods. RB (0.1 mg) was dissolved in 1 mL $H_2O$, and the solution was added into $V_2C$-PVA (10 mg mL$^{-1}$, 9 mL) dispersion stirring away from light for 12 h at 4 °C. After dialysis to remove the free dye, the formed RB-$V_2C$ was kept at 4 °C for further use. Both L929 and PC 12 cells were placed into 20 mm glass-bottom cell culture dishes ($1 \times 10^5$ cells per dish) and preincubated with endocytic selective inhibitors such as amiloride (5 mM), chlorpromazine (5 µM), and nystatin (125 µM) for 1 h at 37 °C. After incubated with RB-$V_2C$ (400 µg mL$^{-1}$) for 4 h, the washed cell samples were further stained with DAPI for 5 min and observed by using CLSM. It is noted that chlorpromazine can inhibit clathrin-mediated endocytosis. Nystatin inhibits caveolae/lipid-mediated endocytosis by decomposing cholesterol. Amiloride restrains micropinocytosis.

**Assessment of protein carbonylation**. The levels of protein carbonylation were detected by spectrophotometric assay of 2,4-dinitrophenylhydrazine (DNPH) derivatives. For the measurement, the cells were initially treated with a protocol similar to that followed for ROS assessment. After treatment, the cells were harvested and lysed. Protein was added to 800 µL of 10 mM DNPH or 800 µL of 2.5 M HCl, followed by incubation in dark at room temperature for 1 h. Totally, 20% (w/v) TCA was then added to the reaction mixture and the protein was pelleted by centrifugation at $23,469 \times g$ for 10 min at 4 °C. Subsequently, the supernatant was removed from each sample and the pellet was rinsed three times with a 1:1 mixture of ethanol/ethyl acetate. Thereafter, the pellet was resuspended in 500 µL of 6 M guanidine hydrochloride solution and centrifuged at $23,469 \times g$ for another 10 min. Finally, the absorbance of the supernatant was measured at 370 nm ($n = 3$ for each group).

**Assessment of lipid peroxidation**. Cellular lipid peroxidation was evaluated by using C11-BODIPY$^{581/591}$ dye. The cells were treated with a protocol similar to as-above mentioned for the measurement of ROS generation. Post treatment, the cells were loaded with 1 mL of 10 µM C11-BODIPY$^{581/591}$ dye for 30 min at 37 °C. The excitation wavelength of C11-BODIPY$^{581/591}$ was set at 488 nm (oxidized form) and 563 nm (nonoxidized form), and fluorescence emission spectra detected with two bandpasses, in the range of 505–550 nm and 570–630 nm. Oxidation of C11-BODIPY$^{581/591}$ was revealed by the change of BODIPY fluorescence from red to green, followed by fluorescence imaging were acquired under a CLSM.

**Assessment of DNA damage**. Post described above treatment, cells were fixed with 500 µL of 4% paraformaldehyde for 20 min at room temperature, and washed with PBS, followed by permeabilization with 500 µL of 0.5% Triton X-100. After rinsing, cells were blocked with 1 mL of 1% BSA in PBS for 1 h and further labeled with the mouse monoclonal anti-phospho-histone-H2AX primary antibody at a dilution of 1:500 overnight at 4 °C. Subsequently, cells were then incubated in the dark with anti-mouse Alexa Fluor 488 secondary antibody at a dilution of 1:500 in 1 mL PBS containing 1% BSA for 1 h. Afterward, the unbounded antibody solution was removed by PBS washing. Finally, cell nuclei were stained with 200 µL DAPI (10 µg mL$^{-1}$) for 5 min and imaged using a CLSM.

**Caspase-3/7 activity analysis**. The intracellular activity of caspase-3/7 was assessed to determine the induction of apoptosis using the caspase 3/7 fluorescent real-time detection kit following the manufacturer's instructions. After different treatment, cells were washed three times with PBS and treated with 200 µL caspase-3/7 work solutions (5 µM) in PBS containing 5% FBS at 37 °C for at least 30 min. All samples were examined and photographed by using CLSM. The excitation/emission maxima for the Caspase-3/7 Green Detection Reagent is 502/530 nm.

**Western blot analysis.** After treatment, cells were collected and washed twice with ice-cold PBS, followed by resuspension in RIPA buffer containing the protease inhibitor to prepare the cell lysates. After incubation on ice for 20 min, debris was removed by centrifugation ($20{,}627 \times g$, 10 min) at 4 °C. Equal loading of protein cell extracts was diluted in sample buffer and subjected to 10% sodium dodecyl sulfate-polyacrylamide gel electrophoresis (SDS-PAGE) and transferred to polyvinyl difluoride membranes. Subsequently, nonspecific binding was blocked with 5% (w/v) non-fat dry milk in Tris-buffered saline containing 0.1% Tween-20 (TBST) for 1 h at room temperature. The samples were immunoblotted with the following primary antibodies: caspase-3 (1:1000 dilution), cleaved caspase-3 (1:1000 dilution), and GAPDH (1:1000 dilution) at 4 °C overnight. After washing, the blots were incubated with the corresponding fluorescent secondary antibody (1:5000 dilution) for 1 h at room temperature. They were further rinsed with TBST, and the target proteins were detected using the Odyssey Infrared Imaging System (LI-COR Biosciences, Lincoln, NE, United States). The level of GAPDH immunoreactivity was used as a control to monitor equal protein loading.

**Animals and treatment.** Six-week-old healthy female Kunming mice and C57BL/6 mice with bodyweight around 18 g were purchased from Slac Laboratory Animal Co., Ltd. (Shanghai, China). The animals were housed in stainless steel and ventilated cages under the standard conditions (temperature: $25 \pm 2$ °C, relative humidity: $60 \pm 10\%$, and light: 12 h light/dark cycle) for seven days prior to treatment. Animals were randomly assigned to test groups and fed sterilized water and pellet food ad libitum. All animal procedures were carried out in strict compliance with the guidelines of the Regional Ethics Committee for Animal Experiments approved by the Administrative Committee of Laboratory Animals of Shanghai Tenth People's Hospital, Tongji University School of Medicine.

**In vivo toxicity assay.** After acclimation, thirty female Kunming mice were randomly assigned to two groups for in vivo toxicity test (12 active mice in each group and 3 backup ones). For Group 1 as the control group, saline solution was utilized as a blank vehicle; Group 2 as the experimental group was intravenously administrated with $V_2C$ at a dose of 15 mg kg$^{-1}$. All groups were intravenously injected once a week.

For histology investigation, the above-mentioned mice were sacrificed and the major organs were harvested from each group at predetermined time-points. Subsequently, all the organs specimens were immediately fixed in 10% formalin, embedded in paraffin, sliced into 4 μm thickness, subjected to H&E staining and finally examined under an inverted phase-contrast microscope ($n = 3$ for each group).

For blood analysis, the hematological and blood biochemical indexes were detected once a week. The whole blood, approximately 1 mL/sample with anticoagulation treatment, was harvested via orbital puncture from each animal. The routine blood test, including the erythrocyte count, white blood cell count, and platelet count were performed by using automated hematology analyzing machine. The serum was obtained by centrifugation of the whole blood at $1{,}467 \times g$ for 15 min. After that, the serum biochemical assays, including alanine aminotransferase, aspartate aminotransferase, blood urea nitrogen, blood protein, creatinine, albumin, globulin, as well as the ratio of albumin to globulin (A/G) were conducted by using a Hitachi 7020 automatic biochemical analyzer ($n = 3$ for each group).

**Ear inflammation models construction and treatment.** Female Kunming mice were randomly assigned to four groups for ear inflammation model construction and treatment. After removing female Kunming mice hair, ear inflammation was induced by topical application of 20 μL of 100 μg mL$^{-1}$ PMA acetone solution on the right ear of each mouse. After 6 h PMA application, inflammation was clinically observed as edema and redness on the ear. Subsequently, mice were anesthetized and subcutaneously administered with $V_2C$ MXenzyme dispersion, followed by injection of 50 μL of 1 mM DCFH-DA in a similar way. After further 30 min treatment, the whole-body photoluminescence imaging was acquired on a PerkinElmer IVIS Lumina Series III imaging system (the excitation filters of 420, 440, 460, 480, 500, 520, 540, 560, 580, 600, 620, 640, 660, 680, 700, 720, 740, 760, and 780 nm, and the emission filters of 520, 570, 620, 670, 710, 790, and 845 nm) using an excitation wavelength of 480 nm and an emission wavelength of 520 nm. Fluorescent quantitation was assessed by using the Living Image In Vivo Imaging Software. Furthermore, multispectral unmixing with Compute Pure Spectrum technology was used to enable accurate autofluorescence removal. For the control group, PBS was injected subcutaneously into the right ear of mice. For 6 h later, mice were sacrificed and the ear skin was processed for histology analysis ($n = 3$ for each group).

**Ankle inflammation models construction and treatment.** Female Kunming mice were randomly assigned for ankle inflammation model construction and treatment. The model of ankle inflammation was established by topical injection of 20 μL of 2 mg mL$^{-1}$ LPS in PBS solution into each ankle of female Kunming mice. After 5 h challenge with LPS, $V_2C$ MXenzyme suspension was injected into one of the inflamed ankles, while the other ankle treated with normal saline solution was regarded as control. At 30 min post injection, 50 μL of 1 mM DCFH-DA was

injected in a similar way. To test the therapeutic effect, the fluorescence images of the specimens were then captured with an IVIS imaging system with an excitation wavelength of 488 nm and an emission wavelength of 520 nm. The resulting fluorescence intensity of each sample was measured by using the Living Image In Vivo Imaging Software. Four hours later, mice were sacrificed and the specimens were collected for histology analysis ($n = 3$ for each group).

**PD models construction and treatment.** Eighteen C57BL/6 mice were randomly assigned and treated with 30 mg kg$^{-1}$ MPTP saline solution intraperitoneally once daily for 10 consecutive days, along with daily saline-treated as control. After the PD model was successfully constructed, the mice were randomly divided into three groups including (1) saline + saline, (2) MPTP + saline, and (3) MPTP + $V_2C$ ($n = 6$ for each group). For the stereotaxic surgery, each C57BL/6 mouse was anesthetized with an intraperitoneal injection of chloral hydrate (350 mg kg$^{-1}$) and then placed on a Narishige stereotaxic apparatus (Yuyanbio, China) to achieve a flat skull position (Supplementary Information, Fig. 31). The scalp was incised to expose the parietal bone. After that, a small hole was drilled at the following coordinates: lateral (L) = 2.3 mm, anteroposterior (AP) = 0.8 mm, ventral (V) = −3.5 mm from the bregma point. The $V_2C$ MXenzyme dispersion (10 mg mL$^{-1}$, 4 μL) or sterile saline solution (4 μL) was injected unilaterally into the striatum at the rate of 1 μL min$^{-1}$ and a 5 min lag was allowed before needle extraction to avoid brain damage.

After 2 weeks, the mice were deeply anesthetized with 5% chloride hydrate. The obtained 18 brains (6 per group) were fixed with 4% paraformaldehyde and embedded in paraffin. Brains in each group were randomly assigned to two subgroups: three brains were subsequently cut into 4-μm-thick serial coronal sections and the other three were serially cut into 4-μm-thick transverse sections using a pathology slicer (Leica RM2016, Germany). For immunohistochemical staining, these sections were put into xylene three times for 15 min to remove paraffin, rehydrated via a gradient of ethanol (100%, 100%, 95% and 70% ethanol) for 5 min in each concentration, and then rinsed using DI water. After that, they were brought to a boil in sodium citrate buffer (10 mM, pH 6.0) for antigen retrieval in a microwave oven (Galanz P70D20TL-P4, China), maintained at just below the boiling temperature for 8 min, cooled to room temperature, placed in PBS (pH 7.4) and shaken on the decolorization shaker (Servicebio TSY-B, China) 3 times for 5 min each. Subsequently, the samples were placed in 3% $H_2O_2$ for 25 min and washed with PBS three times for 5 min each. To prevent antibody nonspecific binding, 3% BSA solution was added to cover the samples for 30 min at room temperature. After removing the blocking solution, the sections were incubated with anti-IBA-1 antibody (1:100 dilution), anti-TH antibody (1:200 dilution), or anti-4-HNE antibody (1:100 dilution) at 4 °C overnight, and then washed by shaking 3 times for 5 min each. After covering with secondary horseradish-peroxidase-conjugated anti-rabbit IgG antibody (1:500 dilution), they were incubated for 50 min at room temperature and washed 3 times for 5 min each. Diaminobenzidine (DAB) color developing solution as newly prepared was added and 5 min generally provided an acceptable staining intensity, which could be controlled under the microscope (Nikon E100, Japan). Next, the sections were counterstained with hematoxylin for 3 min, washed with water, and subsequently located in 75% ethanol for 5 min, 85% ethanol for 5 min, 100% ethanol for 5 min,100% ethanol for 5 min, n-butanol for 5 min and xylene for 5 min. Finally, the samples were mounted onto gelatine-coated microscope slides with coverslips using the mounting medium, and the slides were visualized on a microscope (Nikon DS-U3, Japan).

For immunofluorescence staining, these sections were put into xylene two times for 15 min to remove paraffin, rehydrated via a gradient of ethanol (100%, 100%, 95%, and 70% ethanol) for 5 min in each concentration, and then rinsed using DI water. After that, the sections were brought a boil in ethylene diamine tetraacetic acid buffer (1 mM, pH 8.0) for antigen retrieval, maintained at just below the boiling temperature for 8 min, cooled to room temperature, and washed with PBS (pH 7.4) 3 times for 5 min each. Subsequently, the liquid was removed and the samples were marked using a hydrophobic pen. In order to prevent nonspecific binding, 3% BSA solution was added to cover the marked tissue for 30 min, followed by removing the blocking solution slightly. The slides were incubated with primary antibody overnight at 4 °C and then washed with PBS (pH 7.4) 3 times for 5 min each, followed by incubation with Alexa Fluor 488 conjugated anti-rabbit IgG antibody (1:200 dilution), Alexa Fluor 594 conjugated anti-rabbit IgG antibody (1:200 dilution), and Alexa Fluor 647 conjugated anti-rabbit IgG antibody (1:200 dilution) at room temperature for 50 min in dark condition. After rinsing three times with PBS (pH 7.4) for 5 min each, the samples were treated with DAPI solution for 1 min at room temperature and washed 3 times for 5 min each. Spontaneous fluorescence quenching reagent was added and kept for 5 min. Next, the sections were mounted onto the gelatine-coated microscope slides with coverslips using an anti-fade mounting medium.

For the immunohistochemical staining analysis, the cell counting percentage of TH+, IBA-1+, or 4-HNE+ cells was quantified in a blind manner, which was conducted by analyzing the treatment conditions with randomly allocated groups through the unbiased method using an image analysis program Image J (Fiji). Quantification was performed by averaging five fields per section, and five different sections were independently quantified for each mouse. Subsequently, the result was expressed as (experimental group/control group) × 100%. Quantitative

comparisons were carried out on image sets collected under the same imaging and acquisition conditions.

**Statistical analysis**. All quantitative results were shown as mean ± standard deviation. All mean values represent the average of all experimental groups analyzed. Statistically analyzed comparisons were generated by a blinded counter. The statistical significance of differences among groups was carried out by using one-way analysis of variance analysis followed by Tukey's post-test. The tests were regarded as statistically significant at $*p < 0.05$, $**p < 0.01$, and $***p < 0.001$.

**Reporting summary**. Further information on research design is available in the Nature Research Reporting Summary linked to this article.

## Data availability

The authors declare that all data needed to support the finding of this study are presented in the article and the Supplementary Information. Any other data related to this work are available from the corresponding authors upon reasonable request. A reporting summary for this article is available as a Supplementary Information file.

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

## Acknowledgements
This work was financially supported by the National Key R&D Program of China (Grant no. 2016YFA0203700), National Science Foundation for Young Scientists of China (Grant no. 51802336), and National Nature Science Foundation of China (Grant nos. 51672303, 51722211, and 52072393).

## Author contributions
W.F. and Y.C. conceived the research. Y.L. and Y.C. supervised the project and commented on the project. W.F. synthesized and characterized the nanomaterials. W.F., H.H., X.H., and L.D. conducted in vitro and in vivo experiments. X.H. carried out the WB experiment. W.F., M.C., and H.X. analyzed the data. W.F. wrote the paper. All authors discussed the experimental procedures and results.

## Competing interests
The authors declare no competing interests.
