## [Peer Review File · Nature Communications]

Reviewers' Comments:

Reviewer #1 (Remarks to the Author):

The work presented by Feng et al. describes the instinct enzyme-mimicking catalytic activity of vanadium carbide (V₂C) MXene. Their results indicated that V₂C MXene could serve as six naturally-occurring enzymes, including superoxide dismutase (SOD), catalase (CAT), peroxidase (POD), glutathione peroxidase (GPx), thiol peroxidase (TPx) and haloperoxidase (HPO). Moreover, the V₂C MXene could be efficiently uptaken by L929 and PC12 cells and protect the cells from ROS-induced damage through a catalytic reaction. The authors also used inflammation and neurodegeneration animal models to evaluate their therapeutic effects. Finally, the underlying mechanism of the enzyme-mimicking activities was proposed based on corresponding characterizations. Overall, the new discovery of V₂C MXene as an artificial enzyme not only supplement the family of nanozymes, but also extend the application of MXenes materials from traditional chemical catalysis and energy storage to biomedicine. The experiments are well designed and the results are convincing. I would recommend its publication in Nature Communication after the following revisions.

1. The author tested the V₂C MXene as positive for six enzyme-mimicking activities, which was exciting and inspiring. As the field of nanozyme developed, the catalytic activity of various natural enzymes can be mimicked by different nanomaterials. Readers may curious about whether V₂C MXene could mimic other natural enzymes. A systematic study of V₂C MXene as nanozyme would be recommended including oxidase, phosphatase, glucose oxide, protease-like activity, and nuclease-like activity.
2. The authors proved the internalization of V₂C MXene was through the endocytic pathway, the endocytic inhibitors were recommended to investigate the uptake mechanism.
3. In the introduction, more pioneering work and progress need to be cited when the author describes the field of nanozyme (e.g. Nature Nanotechnology, 2007, 2, 577; Advanced Materials, 2015, 1097). Besides, the advantages of V₂C MXene over the reported nanozyme were recommended to discuss in the manuscript.
4. In my own opinion, the concept of "MXenzyme" is good but not adequate for the present unless there are enough researches to demonstrate the MXene as nanozyme.

Minor points:

1. The words and expressions in the scientific paper should be highly clear, accurate and concise. The following tips may be helpful.

Line 24 "ROS exacerbation-mediate" is not easy to understand;

Line 31 "the highly chemical-reactive and oxygen-involved substances and free radicals" may be condensed to "chemically reactive molecules containing oxygen";

Line 33 "role in adjusting cell functions" may be specified as "role in cell signaling and homeostasis".

Line 56 "very few vanadium-based nanozymes" should be cited by the corresponding papers.

2. Some words in Scheme 1 are not readable. Please revise this.
3. The contrast of some words in Figure 2 needs to be improved such as “k, r, s, t”.
4. In Figure 4, the font and line should be unified such as “the line color of cell viability and ROS intensity”. Besides, the author should also mark the meaning of “L/D”.
5. In Figure 4k and 4j, the labels are confusing for the existence of two V2C. We would recommend using “V2C + H2O2” and “V2C + H2O2 +BSO” for better understanding.
6. For Western blot, the same background (brightness, contrast) is recommended.
7. In Figure 6j, more labels are needed to mark the horizontal ones.
8. In Methods, the full names of some chemicals are needed (e.g. TPAOH, BSO, HF, MPTP, PVA).

Reviewer #2 (Remarks to the Author):

In this manuscript, the authors reported a two-dimensional (2D) vanadium carbide (V2C) MXene nanoenzyme for ROS elimination and antioxidative treatment. This V2C MXene nanoenzyme could mimic various redox enzymes including SOD, CAT, POD, GPx, TPx and HPO. Both in vitro and in vivo experiments demonstrated the ROS scavenging ability of this V2C MXene nanoenzyme. The exact mechanism for ROS scavenging (including catalyzing O₂⁻ into H₂O₂ and O₂, decomposing H₂O₂ into O₂ and H₂O, and eliminating •OH) was also studied. Moreover, this V2C MXene nanoenzyme had the potential for the treatment of ROS exacerbation-mediated inflammatory and neurodegenerative diseases. The material is new and the application is innovative. This definitely represents one of the best work in the field of nanozymes. The whole manuscript is well organized. Thus, this manuscript was recommended for publication after addressing the following minor concerns.

1. In Figure 2, the element mapping (V, Al and C element signals) of ML V2C MXenes and FL V2C MXenes should also be provided to demonstrate the successful etching of Al layers and the distribution of V and C elements.
2. The POD-like activity of the V2C MXenes is similar to GPx, and both of them need the reductant (GSH or TMB) to catalyze the reduction of H₂O₂. Thus, the schematic illustration in Figure 3e should be revised. The POD-like activity cannot directly catalyze H₂O₂ into H₂O.
3. The V2C MXenes exhibited a good antioxidative capability. The authors can consider providing the comparison of the V2C MXenes with the widely applied antioxidants (for example vitamin C) in vitro. Or performance parameters of vitamin C can be extracted from literatures for comparison.
4. The confocal images in Figure 5f didn't show an obvious decrease of the blue fluorescence in the V2C MXenes groups. Please provide a more convincing result.
5. The resolution of the images in Figure 6d and 6g is too low to distinguish the infiltrating inflammatory cells. Please indicate them in the images.
6. What's the mechanism for POD and HPO-like activity of V2C MXenzyme? Please elaborate it in the scheme and main text.

7. Now the references are mainly related to the inorganic materials. Some literatures on organic nanoenzymes can be introduced and cited in the introduction section of this manuscript (J. Am. Chem. Soc., 2019, 141, 4073; Angew. Chem. Inter. Ed., 2018, 57, 3995; Acc. Chem. Res. 2020, DOI: 10.1021/acs.accounts.9b00569).

Reviewer #3 (Remarks to the Author):

The authors propose the use of V2C as a nanoenzyme for cellular protection. While the concept is interesting there are several flaws throughout the paper. These are major concerns and do not warrant publication in its current condition. It is a shame because I began reading with high interest in this novel topic, but this work is unfortunately far from the par expected – particularly in the lack of clarity in the methodology.

Major Points –

The data in figure 3b (catalytic breakdown of H₂O₂) and the accompanying methods do not show any report of the number of repetitions for this study and only one line is shown. Therefore, the main text “The fluorescent intensity displays a significant downtrend” is incorrect. Also replicates should be added to this study.

I can see the idea of all the different assays in Figure 3. It looks impressive that V2C acts in many different mimicking ways, but actually CAT, POD and GPx activity is all driven by the same mechanism surely, which is the break down of H₂O₂. This is the part that the V2C is undertaking, then the key difference between the assays is the substrate addition (either nothing (in which case O₂ is evolved, TMB or GSH). Therefore, would it not be more to the point that V2C catalyzes the breakdown of H₂O₂? Then state that this could have implications for a multitude of cellular activities.

For instance – “apparent TPx-mimicking behavior by catalyzing the reduction of H₂O₂ to water (Supplementary Fig. 7), indicating that V2C MXenzyme could utilize other thiol-molecules without GSH, which considers the factual situation that the GSH homeostasis is commonly disturbed under oxidative stress²⁷.” – Well the authors show breakdown of H₂O₂ without any other substrate (i.e. H₂O and oxygen production), so no wonder that this is also able to used in conjunction with GSH, Br- etc. Perhaps I am missing something, and I apologise if I have, but this seems like many studies based on the same initial key finding, that V2C catalyses the breakdown of H₂O₂.

I could not find the methods for Figure 3h

No numbers of replicates given for the cell viability assay, or any indication what the error bars represent (Figure S12) – Actually, this rings true for all in vitro experiments and must be addressed before publication. Without replicate numbers and transparent methodology, including statistical analysis the work cannot be truly assessed.

Many of the methods are not clear enough to reproduce the results. As just two examples of the many: 1) "After incubation for 24 h, the medium was discarded and exposed to H₂O₂ or H₂O₂/FeSO₄ in the presence of different concentrations from 0 to 200 µg/mL of V2C MXenzyme for a further 12 h. – So the discarded medium was exposed to the H₂O₂ (obviously not), but the cells were? How much medium per well, what concentration of H₂O₂ was used (vital information), how many replicates. And 2) "irradiated under 365-nm UV lamp." What manufacturer, what intensity of UV was used, at what distance from the cell etc etc.

Figure 6j is produced far too small for the reader to interpret any of the findings. Clear representative images of the model, the TH loss, through multiple brain sections are required for the reader to see the impact of the MPTP alone and with V2C.

The analysis of the IBA1 and 4-HNE is clearly done on different brain regions (SI Fig 21 and 22) the +V2C group is consistently at a different anterior/posterior position and was shown to be more medial than the others. This renders the analysis meaningless.

in addition to the above comment – the areas analysed are not even the injection site (stated as the striatum).

No injection coordinates, stereotactic information given, in sum, the PD model data cannot be repeated or properly interpreted and is not fit for publication.

Minor Points

I would recommend removing words such as tremendous, intriguing and admirable etc.

The first sentence of the introduction could be broken down into two to be more clear

would subject biosystem to (would subject a biosystem to)

function inactively under pathological conditions... (become inactive under pathological conditions?)

found in the emerging of a large...

Despite of the high prospect...

Grammatical errors too numerous to list individually. A thorough proof read is required.

Reference required for such a statement "one of the forty essential micronutrients"

Personally, I think that Figure 1 could be greatly improved without the smileys and shield, and instead have a single image of normal cell function – highlighting every area where the V2C can assist.

In general a good hypothesis and clear set of objectives (or an aim) is missing from the introduction.

AI is around 2.60 _ (units not visible in the pdf file).

Figure 3a – Y axis label "inhibitor rate of SOD" is unclear and does not seem to match the main text which stated a drop in the formazan product. I think the aim was to mimic SOD not inhibit it.

Wording in the main text refers the reader to a schematic – not data “spectrophotometrically real-time measuring the decrease of NADPH level at 340 nm (Fig. 3f)”

All the electron microscope characterization was carried out with using the PVA functionalized V2C. Perhaps these studies could be included, since this is the final “product” the cells receive.

No mention of rationale for choosing the cell types used.

“MXenzyme” line 284

Ambiguous subheadings in methods – eg “V2C MXenzyme Protecting H₂O₂ L929 and PC12 Cells from Fenton’s Reagent-Induced Oxidative Stress”

Response to Reviewer # 1

Comments from reviewer # 1:

The work presented by Feng et al. describes the instinct enzyme-mimicking catalytic activity of vanadium carbide (V₂C) MXene. Their results indicated that V₂C MXene could serve as six naturally-occurring enzymes, including superoxide dismutase (SOD), catalase (CAT), peroxidase (POD), glutathione peroxidase (GPx), thiol peroxidase (TPx) and haloperoxidase (HPO). Moreover, the V₂C MXene could be efficiently uptaken by L929 and PC12 cells and protect the cells from ROS-induced damage through a catalytic reaction. The authors also used inflammation and neurodegeneration animal models to evaluate their therapeutic effects. Finally, the underlying mechanism of the enzyme-mimicking activities was proposed based on corresponding characterizations. Overall, the new discovery of V₂C MXene as an artificial enzyme not only supplement the family of nanozymes, but also extend the application of MXenes materials from traditional chemical catalysis and energy storage to biomedicine. The experiments are well designed and the results are convincing. I would recommend its publication in Nature Communication after the following revisions.

Response: Thanks very much for your positive comments and kind recommendation. Please find the following detailed responses to your comments and suggestions.

(1) *The author tested the V₂C MXene as positive for six enzyme-mimicking activities, which was exciting and inspiring. As the field of nanozyme developed, the catalytic activity of various natural enzymes can be mimicked by different nanomaterials. Readers may curious about whether V₂C MXene could mimic other natural enzymes. A systematic study of V₂C MXene as nanozyme would be recommended including oxidase, phosphatase, glucose oxide, protease-like activity, and nuclease-like activity.*

Response: We appreciate very much for your professional suggestion. Based on the reviewer's constructive suggestion, we have conducted more systematic experiments to test whether V₂C MXene could mimic other natural enzymes. The detailed experimental procedure and data are supplemented in Revised Supplementary Information (Page 48,

Supplementary Information). First, we investigated the oxidase (OXD)-like activity, which reduces oxygen into water in the presence of a hydrogen (H) donor. In order to monitor the reaction, 3,3',5,5'-Tetramethylbenzidine (TMB) was introduced as the H donor, as the oxidized TMB develops a blue product with absorbance at 652 nm. V₂C MXene doesn't exhibit the capability to oxidize TMB in sodium acetate (NaAc) buffer (Supplementary Fig. 44a), revealing that it doesn't possess OXD-like activity. Furthermore, V₂C MXene cannot mimic xanthine oxidase (XOD) that catalyzes the sequential oxidation of hypoxanthine to xanthine and xanthine to uric acid and hydrogen peroxide (Supplementary Fig. 44b). Next, we assessed the phosphatase-like activity using para-nitrophenyl phosphate (pNPP), which becomes an intense yellow soluble product para-nitrophenol under alkaline conditions and can be conveniently tested at 405 nm on a spectrophotometer. As shown in Supplementary Fig. 44c, V₂C MXene doesn't catalyze the decomposition of pNPP to p-nitrophenol and phosphate, indicating no obvious phosphatase-like activity. Glucose oxidase (GOD)-like activity is determined by a coupled enzyme assay, in which GOD oxidizes D-glucose for the production of hydrogen peroxide (H₂O₂) that reacts with o-dianisidine, generating a colorimetric (500 nm) product, proportional to GOD amount. Almost no significant absorbance of V₂C MXene with D-glucose is observed at 500 nm, confirming no GOD-like activity for V₂C MXene (Supplementary Fig. 44d). In addition, the protease can hydrolyze casein to produce tyrosine, which can reduce phosphomolybdic acid compound to tungsten blue, possessing a characteristic absorption peak at 680 nm. After the activity test, we find that V₂C MXene doesn't display protease-like activity (Supplementary Fig. 44e). Based on these results, it can be concluded that V₂C MXene doesn't feature OXD-, XOD-, phosphatase-, GOD- and protease-like activity. It is noted that there are thousands of enzymes in the nature and human body, such as lipases, amylase, maltase, trypsin, lactase, acetylcholinesterase, helicase, DNA polymerase and so on. Therefore, we cannot test all of these enzymes in this work. In the following researches, we expect to launch a systematic study of the activities of V₂C MXene-based artificial enzymes, which is strongly based on the practical application conditions and requirements.

(2) *The authors proved the internalization of V₂C MXene was through the endocytic pathway, the endocytic inhibitors were recommended to investigate the uptake mechanism.*

Response: Thanks very much for your constructive suggestion. Based on the reviewer's suggestion, we have investigated the endocytic mechanism of V₂C MXenzyme. Endocytosis is a process involving the uptake of nanomaterials from the outside to the inside of cells. The endocytosis mechanisms mainly include clathrin-mediated endocytosis, caveolae/lipid-mediated endocytosis and micropinocytosis. Pharmacologically selective inhibitors are usually employed to study the mechanism of endocytosis. Among them, chlorpromazine can inhibit clathrin-mediated endocytosis. Nystatin inhibits caveolae/lipid-mediated endocytosis by decomposing cholesterol. Amiloride restrains micropinocytosis. It is found that either amiloride or nystatin has a significant inhibitory effect on endocytosis of Rhodamine B-labelled V₂C MXene, revealing that the uptake of V₂C MXenzyme in both L929 and PC 12 cells involves caveolae/lipid-mediated endocytosis and micropinocytosis. Accordingly, the mechanisms of endocytosis by V₂C MXene have been provided in the Revised Manuscript, which reads "It is found that either amiloride or nystatin has a significant inhibitory effect on endocytosis of Rhodamine B-labelled V₂C MXenzyme, revealing that the cellular uptake of V₂C MXenzyme in both L929 and PC 12 cells involves caveolae/lipid-mediated endocytosis and micropinocytosis (Supplementary Fig. 23)." (Page 9, Revised Manuscript).

"V₂C MXenzyme was labeled by a red fluorescence dye Rhodamine B (RB) by the following methods. RB (0.1 mg) was dissolved in 1 mL H₂O, and the solution was added into V₂C-PVA (10 mg mL⁻¹, 9 mL) dispersion stirring away from light for 12 h at 4 °C. After dialysis to remove free dye, the formed RB-V₂C was kept at 4 °C for further use. Both L929 and PC 12 cells were placed into 20 mm glass-bottom cell culture dishes (1×10⁵ cells per dish) and pre-incubated with endocytic selective inhibitors such as amiloride (5 mM), chlorpromazine (5 μM) and nystatin (125 μM) for 1 h at 37 °C. After incubated with RB-V₂C (400 μg mL⁻¹) for 4 h, the washed cell samples were further stained with DAPI for 5 min and observed by using CLSM (n = 3 for each group). It is noted that chlorpromazine can inhibit clathrin-mediated endocytosis. Nystatin inhibits caveolae/lipid-mediated endocytosis by

decomposing cholesterol. Amiloride restrains micropinocytosis.” (Page 22, Revised Manuscript).

(3) *In the introduction, more pioneering work and progress need to be cited when the author describes the field of nanozyme (e.g. Nature Nanotechnology, 2007, 2, 577; Advanced Materials, 2015, 1097). Besides, the advantages of V₂C MXene over the reported nanozyme were recommended to discuss in the manuscript.*

Response: Thank you very much for the constructive suggestion, which is highly appreciated.

(a) According to the reviewer’s suggestion, we have cited these pioneering works and progresses in the Revised Manuscript (Page 2, Revised Manuscript; Ref. 10, 17).

(b) Considering the relatively small number of atomic layers, the thickness of a single V₂C MXene is several nanometers. The physical properties such as ultrahigh surface area-to-volume ratio enable extensive surface interactions between V₂C MXenes and reactive oxygen species (ROS), leading to high catalytic activity. In addition, in the presence of both complete vanadium atomic layers and plentiful surface functional groups such as hydroxyl, oxygen or fluorine, V₂C MXenes uniquely combine the metallic conductivity of transition metal carbides with the hydrophilic nature, which endows V₂C MXene with attractive electronic properties. Therefore, the as-constructed 2D V₂C MXenzyme conducts six enzyme-mimicking activities under the physiological condition, including SOD, CAT, POD, GPx, TPx and HPO, which is superior as compared to Cu_xO (three enzyme-like activities for CAT, GPx and SOD) (*J. Am. Chem. Soc.* 2019, 141, 2, 1091–1099), Prussian blue (three enzyme-like activities for CAT, POD and SOD) (*J. Am. Chem. Soc.* 2016, 138, 18, 5860–5865), Fe-N/C single-atom (four enzyme-like activities for POD, oxidase, CAT and GPx) (*Chem. Commun.*, 2019, 55, 14534-14537) and N-doped porous carbon nanospheres (four enzyme-like activities for oxidase, POD, CAT and SOD) (*Nat. Commun.*, 2018, 9, 1440), demonstrating that 2D V₂C MXene holds a broad-spectrum ROS removal capacity. According to the reviewer’s suggestion, the specific advantages and features of V₂C MXene over the reported nanozyme in the literatures have been provided in Revised Manuscript, which is expressed as “ ...which is superior as compared to Cu_xO (three enzyme-like

activities for CAT, GPx and SOD)⁶, Prussian blue (three enzyme-like activities for CAT, POD and SOD)³², Fe-N/C single-atom (four enzyme-like activities for POD, oxidase, CAT and GPx)³⁵ and N-doped porous carbon nanospheres (four enzyme-like activities for oxidase, POD, CAT and SOD)¹¹, demonstrating that 2D V₂C MXenzyme holds a broad-spectrum ROS removal capacity.” (Page 8, Revised Manuscript).

(4) *In my own opinion, the concept of “MXenzyme” is good but not adequate for the present unless there are enough researches to demonstrate the MXene as nanozyme.*

Response: Thank you very much for the kind suggestion, which is highly appreciated. We totally understand reviewer’s concern. In early 1960s, Ronald Breslow and co-workers found cyclodextrins could synthesize amino acids, cleave bonds and even work in a similar way to vitamin B1, and then they proposed the term of “artificial enzyme” for enzyme mimics, which is a very important and exciting branch of biomimetic chemistry inspired by nature and aims to imitate the essential and general principles of natural enzymes using alternative materials. In 1971, Irving M. Klotz and co-workers discovered that substitution of the primary amines of polyethyleneimine with suitable nucleophilic groups creates a polymer with remarkable catalytic properties in the actual hydrolysis of nitrophenyl esters to acid and phenol, and coined the term “Synzyme”, deriving from synthetic enzyme. In 2004, Scrimin, Pasquato and co-workers used triazacyclonane-functionalized gold nanoparticles as catalysts for transphosphorylation reaction, and the facile synthesis and the outstanding catalytic properties prompted them to coin “nanozymes” in analogy to the nomenclature of synzymes. We have been engaged in the research field of 2D biomaterials (*e.g.*, MXenes) based biomedicine for years, spanning from the development of synthetic methodologies for the fabrication of novel 2D biomaterials, to their corresponding extensive biomedical applications. During this time period, we have made substantial progresses in this field by bringing readers with a series of publications. For instance, in 2019, we developed 2D niobium carbide (Nb₂C) MXene as a radioprotectant and explored its application in scavenging free radicals against ionizing radiation (ACS Nano, 2019, 13(6), 6438-6454). In this study, we fabricated and found that a 2D vanadium carbide (V₂C) MXene nanoenzyme

can mimic up to six naturally-occurring enzymes, including superoxide dismutase (SOD), catalase (CAT), peroxidase (POD), glutathione peroxidase (GPx), thiol peroxidase (TPx) and haloperoxidase (HPO). They encouraged us to extensively promote the progress in the field of MXene-based enzyme mimics. It is highly expected that 2D MXene would lead to a burgeoning research interest in applications of artificial catalysts mimicking the natural enzyme functions. Based on the above consideration, we herein would like to remain the term of MXenzyme in this work and expect the emerging of versatile 2D MXene-based nanoenzymes based on the large family members of MAX-based ceramics and 2D MXenes with versatile chemical compositions. Thank you very much again for your kind suggestion.

(5) The words and expressions in the scientific paper should be highly clear, accurate and concise. The following tips may be helpful. Line 24 “ROS exacerbation-mediate” is not easy to understand; Line 31 “the highly chemical-reactive and oxygen-involved substances and free radicals” may be condensed to “chemically reactive molecules containing oxygen”; Line 33 “role in adjusting cell functions” may be specified as “role in cell signaling and homeostasis”. Line 56 “very few vanadium-based nanozymes” should be cited by the corresponding papers.

Response: We appreciate very much for your helpful suggestions and kind reminding. Based on the reviewer’s constructive suggestions, we have checked and corrected these words and expressions in the Revised Manuscript. For example,

“ROS exacerbation-mediated” has been changed to “ROS-mediated” (Page 1, Revised Manuscript).

“the highly chemical-reactive and oxygen-involved substances and free radicals” has been condensed to “chemically reactive molecules containing oxygen” (Page 2, Revised Manuscript).

“role in adjusting cell functions” has been specified as “role in cell signaling and homeostasis” (Page 2, Revised Manuscript).

The corresponding references have been added for “very few vanadium-based nanozymes” (Page 2, Revised Manuscript; Ref. 24, 25).

(6) *Some words in Scheme 1 are not readable. Please revise this.*

Response: Thank you very much for pointing this issue out, which is highly appreciated. We have revised Scheme 1 to make it more readable accordingly (Page 3, Revised Manuscript).

(7) *The contrast of some words in Figure 2 needs to be improved such as “k, r, s, t”.*

Response: Thank you very much for pointing this out. We have improved the contrast of words such as “k, r, s and t” in Fig. 2 accordingly (Page 5, Revised Manuscript).

(8) *In Figure 4, the font and line should be unified such as “the line color of cell viability and ROS intensity”. Besides, the author should also mark the meaning of “L/D”.*

Response: Thank you very much for the kind reminding and suggestion. We have unified the font and line in Fig. 4, and have marked the meaning of “L/D” for “Live and dead” accordingly (Page 10, Revised Manuscript).

(9) *In Figure 4k and 4j, the labels are confusing for the existence of two V_2C . We would recommend using “ $V_2C + H_2O_2$ ” and “ $V_2C + H_2O_2 + BSO$ ” for better understanding.*

Response: Thank you very much for pointing this issue out. We have corrected the labels of Fig. 4k and 4j accordingly (Page 10, Revised Manuscript).

(10) For Western blot, the same background (brightness, contrast) is recommended.

Response: Thanks very much for your suggestion. We have adjusted the background of Western blot in Fig. 5b accordingly (Page 12, Revised Manuscript).

(11) In Figure 6j, more labels are needed to mark the horizontal ones.

Response: Thanks very much for your kind suggestion. We have added the horizontal labels in Fig. 6j (Page 13, Revised Manuscript).

(12) In Methods, the full names of some chemicals are needed (*e.g.* TPAOH, BSO, HF, MPTP, PVA).

Response: Thank you very much for pointing this out. The full names of the chemicals, including tetrapropylammonium hydroxide (TPAOH), buthionine sulfoximine (BSO), hydrofluoric acid (HF), 1-methyl-4-phenyl-1,2,3,6-tetrahydropyridine (MPTP), polyvinyl alcohol (PVA), phorbol 12-myristate 13-acetate (PMA), lipopolysaccharide (LPS), hematoxylin and eosin (H&E), reactive oxygen species (ROS) assay kit, glutathione peroxidase (GPx) assay kit, total antioxidant capacity (TAC) assay kit, cell counting kit-8 (CCK-8) assay kit, and glutathione (GSH)/oxidized glutathione (GSSG) assay kit, have been provided in the section of Methods (Page 17, Revised Manuscript).

Response to Reviewer # 2

Comments from reviewer # 2:

In this manuscript, the authors reported a two-dimensional (2D) vanadium carbide (V₂C) MXene nanoenzyme for ROS elimination and antioxidative treatment. This V₂C MXene nanoenzyme could mimic various redox enzymes including SOD, CAT, POD, GPx, TPx and HPO. Both in vitro and in vivo experiments demonstrated the ROS scavenging ability of this V₂C MXene nanoenzyme. The exact mechanism for ROS scavenging (including catalyzing O₂⁻ into H₂O₂ and O₂, decomposing H₂O₂ into O₂ and H₂O, and eliminating •OH) was also studied. Moreover, this V₂C MXene nanoenzyme had the potential for the treatment of ROS exacerbation-mediated inflammatory and neurodegenerative diseases. The material is new and the application is innovative. This definitely represents one of the best work in the field of nanozymes. The whole manuscript is well organized. Thus, this manuscript was recommended for publication after addressing the following minor concerns.

Response: Thanks very much for your positive comment and kind recommendation. Please find the following detailed responses to your comments and suggestions.

(1) *In Figure 2, the element mapping (V, Al and C element signals) of ML V₂C MXenes and FL V₂C MXenes should also be provided to demonstrate the successful etching of Al layers and the distribution of V and C elements.*

Response: Thank you very much for the constructive suggestion. Based on the reviewer's suggestion, we have provided the data of scanning transmission electron microscopy (STEM), scanning electron microscopy (SEM), energy-dispersive X-ray spectroscopy (EDX) and corresponding elemental analysis for the multilayered and single- or few-layered V₂C MXenes, which have been provided in Revised Supplementary Information (Supplementary Table 1-3, Supplementary Fig. 5, 6 and 8), which reads

“... EDX element mapping (Supplementary Fig. 5a and 6a) and corresponding elemental analysis (Supplementary Fig. 5b, 6b and Supplementary Table 2) prove that almost no Al layer is remaining and the region is fully transformed into MXenes.” (Page 5, Revised

Manuscript).

“The STEM images with EDS mapping results confirmed that Al element has been selectively etched, and V, C, F and O atoms are uniformly distributed throughout the entire nanosheets, where F and O atoms originate from the introduced groups of -F, -O and -OH (Supplementary Fig.8, Supplementary Table 3).” (Page 6, Revised Manuscript).

(2) *The POD-like activity of the V₂C MXenes is similar to GPx, and both of them need the reductant (GSH or TMB) to catalyze the reduction of H₂O₂. Thus, the schematic illustration in Figure 3e should be revised. The POD-like activity cannot directly catalyze H₂O₂ into H₂O.*

Response: Thank you very much for pointing this issue out, which is highly appreciated. According to the reviewer’s comment, we have revised the schematic illustration and added TMB in Fig. 3e (Page 7, Revised Manuscript).

(3) *The V₂C MXenes exhibited a good antioxidative capability. The authors can consider providing the comparison of the V₂C MXenes with the widely applied antioxidants (for example vitamin C) in vitro. Or performance parameters of vitamin C can be extracted from literatures for comparison.*

Response: Thanks very much for your kind suggestion. In this work, in order to assess the ROS-scavenging activity of V₂C MXenzyme and predict its potential for oxidation resistance, the total antioxidant capacity (TAC), an important parameter for assessing the antioxidant potentials, is expressed as Trolox equivalents, which is a water-soluble vitamin E analog. The results reveal that TAC value escalates with the elevated concentrations of V₂C MXenzyme (Fig. 3g). Based on the reviewer’s constructive suggestion, we further employ vitamin C serving as another antioxidant to evaluate the antioxidative capability in the Revised Manuscript (Page 8, Revised Manuscript) and Revised Supplementary Information (Supplementary Fig. 15, Page 19, Revised Supplementary Information). As expected, the TAC of V₂C MXenzyme increased significantly with the elevated concentration. At a dosage of 238 μg mL⁻¹, the TAC test result of V₂C MXenzyme is 36 μM vitamin C equivalents.

(4). *The confocal images in Figure 5f didn't show an obvious decrease of the blue fluorescence in the V₂C MXenes groups. Please provide a more convincing result.*

Response: Thank you very much for pointing out this issue. ROS can induce several types of DNA damage, among which DNA double strand breaks can be potentially lethal to the cells (*Nat. Commun.*, 2018, 9, 2736). In this study, we analyzed the histone H2AX phosphorylated on Ser 139 (γ -H2AX) expression using immunofluorescence staining (green signal not blue), which is a sensitive marker of DNA double strand breaks. Diamidino-2-phenylindole (DAPI) is a blue fluorescent probe that fluoresces selectively and brightly for nuclei location. Remarkably, an increment in γ -H2AX foci formation (green fluorescence) was monitored in the V₂C MXenzyme-untreated groups including H₂O₂, UV irradiation and Fenton's reagent, whereas an apparent decrease was observed in the cells pretreated with V₂C MXenzyme, demonstrating that V₂C MXenzyme confers a protective effect on DNA integrity (Fig. 5f, Page 12, Revised Manuscript; Supplementary Fig. 27, Page 31, Revised Supplementary Information).

(5). *The resolution of the images in Figure 6d and 6g is too low to distinguish the infiltrating inflammatory cells. Please indicate them in the images.*

Response: Thank you very much for pointing this issue out. According to the reviewer's constructive suggestion, we have added red arrows to indicate the infiltrating inflammatory cells (Fig. 6d and 6h) in the Revised Manuscript (Page 13, Revised Manuscript).

(6). *What's the mechanism for POD and HPO-like activity of V₂C MXenzyme? Please elaborate it in the scheme and main text.*

Response: Thank you very much for the constructive question and suggestion. According to the reviewer's suggestion, the underlying POD and HPO-like catalytic mechanisms mediated by the V₂C MXenzyme have been proposed after analyzing the related experimental results based on the V₂C MXene structure in the Revised Manuscript, which is expressed as:

“Finally, for the underlying mechanism of V₂C MXenzyme as POD mimetics, in V₂C

MXene structural model, the V atom is supposed to act as Lewis acid site, but the bond pair of electrons for bridging oxygen atoms behave as Lewis base sites, in which the nucleophilic addition reactions of oxygen happen. Subsequently, the V₂C MXene is supposed to react with H₂O₂ to form an intermediate V-peroxo species (Supplementary Fig. 43), and then the TMB substrate binds to the V-peroxo complexes *via* nucleophilic attack, thus allowing the oxidation reaction of TMB to form the TMB^{*+} species. Because H₂O₂ is a two-electron oxidant, another TMB molecule is required for V₂C MXenzyme regeneration inducing TMBox formation.” (Page 17, Revised Manuscript).

“It is noted that the GSOH produced by this pathway is likely the migration of HOBr from an intermediate of V-OBr in vanadium-dependent haloperoxidase^{17,30}(Supplementary Fig. 42).” (Page 16, Revised Manuscript).

(7). Now the references are mainly related to the inorganic materials. Some literatures on organic nanoenzymes can be introduced and cited in the introduction section of this manuscript (J. Am. Chem. Soc., 2019, 141, 4073; Angew. Chem. Inter. Ed., 2018, 57, 3995; Acc. Chem. Res. 2020, DOI: 10.1021/acs.accounts.9b00569).

Response: Thank you very much for the kind suggestion. According to the reviewer’s constructive suggestion, we have cited these important references on organic nanoenzyme in the introduction section of Revised Manuscript (Page 2, Revised Manuscript; Ref. 18-20).

Response to Reviewer # 3

Comments from reviewer # 3:

The authors propose the use of V₂C as a nanoenzyme for cellular protection. While the concept is interesting there are several flaws throughout the paper. These are major concerns and do not warrant publication in its current condition. It is a shame because I began reading with high interest in this novel topic, but this work is unfortunately far from the par expected – particularly in the lack of clarity in the methodology.

Response: Thanks very much for your constructive suggestions and critical comments, which are highly helpful and important for further improving the manuscript quality. We have conducted more systematic experiments with the addition of more detailed clarification on the methodology of this work. Please find the following detailed responses to your kind comments and suggestions.

(1). *The data in figure 3b (catalytic breakdown of H₂O₂) and the accompanying methods do not show any report of the number of repetitions for this study and only one line is shown. Therefore, the main text “The fluorescent intensity displays a significant downtrend” is incorrect. Also replicates should be added to this study.*

Response: Thank you very much for pointing this issue out. According to the reviewer’s constructive suggestion, we have supplemented three replicates and corresponding values in the Revised Manuscript and Revised Supplementary Information. (Fig. 3b, Page 7, Revised Manuscript; Supplementary Fig. 9, Page 13, Revised Supplementary Information).

(2). *I can see the idea of all the different assays in Figure 3. It looks impressive that V₂C acts in many different mimicking ways, but actually CAT, POD and GPx activity is all driven by the same mechanism surely, which is the break down of H₂O₂. This is the part that the V₂C is undertaking, then the key difference between the assays is the substrate addition (either nothing (in which case O₂ is evolved, TMB or GSH). Therefore, would it not be more to the point that V₂C catalyzes the breakdown of H₂O₂? Then state that this could have implications*

for a multitude of cellular activities. For instance – “apparent TPx-mimicking behavior by catalyzing the reduction of H₂O₂ to water (Supplementary Fig. 7), indicating that V₂C MXenzyme could utilize other thiol-molecules without GSH, which considers the factual situation that the GSH homeostasis is commonly disturbed under oxidative stress²⁷.” – Well the authors show breakdown of H₂O₂ without any other substrate (i.e. H₂O and oxygen production), so no wonder that this is also able to use in conjunction with GSH, Br- etc. Perhaps I am missing something, and I apologise if I have, but this seems like many studies based on the same initial key finding, that V₂C catalyses the breakdown of H₂O₂.

Response: Thank you very much for the kind and professional comments. In this work, we report a V₂C MXenzyme that can mimic SOD, CAT, POD, GPx, TPx and HPO. Among them, it appears that the activity of CAT, POD, GPx, TPx and HPO is all driven by the similar mechanism for breaking down of H₂O₂. Actually, they are different, especially for CAT. We have systematically analyzed the mechanism of each enzyme-mimicking activity in the Revised Manuscript (Fig.7, Page 15, Revised Manuscript) and Revised Supplementary Information (Supplementary Fig. 42 and 43, Page 46 and 47, Revised Supplementary Information).

In addition, H₂O₂, as the stable product of superoxide radical's dismutation *via* superoxide dismutase (SOD), plays a dual role in biological systems. Though H₂O₂ itself is stable and less active, it can be converted into highly active and detrimental hydroxyl radical through Fenton chemistry. Catalase (CAT) is employed as the most efficient enzyme for the conversion of H₂O₂ to less active oxygen. Furthermore, the enzymatic activities are usually pH dependent, with CAT activity dominant at neutral conditions, while the peroxidase (POD) activity was dominant under acid conditions (Supplementary Fig. 11c, Page 15, Revised Supplementary Information), suggesting a different mechanism for the enzyme-mimicking activity. When the V₂C MXenzyme is entrapped within the cytosol, they can convert H₂O₂ into harmless product *via* CAT-mimicking activity. If the V₂C MXenzyme is entrapped into acidic lysosomes, it functions as POD. It is important to develop nanoenzymes that can mimic the function of different cellular antioxidant enzymes (SOD, CAT, GPx and TPx) as these enzymes have differential affinity towards ROS. For H₂O₂, CAT is responsible for

removal of H₂O₂ at high concentration during oxidative stress, while GPx is known to facilitate the concentration of H₂O₂ for cell signaling (*Angew. Chem. Int. Ed.* 2017, 56, 14267-14271; *J. Neurosci.*, 2004, 24 (7): 1531-1540). It has been reported that CAT-GPx cooperativity is important for the proper control of H₂O₂ levels under pathophysiological conditions. Moreover, the V₂C MXenzyme exhibited a general thiol peroxidase-mimicking activity, catalyzing the reduction of H₂O₂ by other thiols such as cysteamine. Furthermore, the GPx-mimicking activities of V₂C MXenzymes were not affected by vanadium haloperoxidase substrates, attributable to the stronger binding affinity and nucleophilic ability of GSH than halides towards the nanozymes. In this study, we used H₂O₂ as a substrate to illuminate V₂C MXenzyme with excellent multifunctional enzyme-mimicking activities. In fact, GPx and TPx can catalytically reduce and scavenge a broad spectrum of hydroperoxides, such as peroxyxynitrite, with the concomitant oxidation of two thiol groups to a disulfide. According to the reviewer's comments, we have further emphasized the V₂C-catalyzed decomposition of H₂O₂ in the Revised Manuscript and comprehensively analyzed the mechanism of V₂C MXenzyme activities, which reads "CAT or CAT mimics have a well-documented capability to catalyze the decomposition of two molecules of H₂O₂ for yielding O₂ and H₂O, therefore preventing the accumulation of H₂O₂ and protecting living organisms from oxidative damage by peroxide³¹." (Page 6, Revised Manuscript).

(3). *I could not find the methods for Figure 3h*

Response: Thank you very much for pointing this issue out, which is highly appreciated. According to the reviewer's kind reminding, we have supplemented the detailed methods for Fig. 3h in the revised manuscript, which reads "Ultraviolet radiation-inhibition activity. 0.8 mM of TMB solutions were placed in 1.5 mL Eppendorf tubes containing V₂C MXenzyme with different concentrations (0, 16, 32, 62.5, 125 and 250 μg mL⁻¹) and irradiation was carried out under a 365 nm ultraviolet lamp (Lichen, China) in a distance of 10 cm at a dose of 20 J cm⁻² for 30 min. After that, the absorption changes of reaction mixture were spectrophotometrically monitored at 652 nm using UV-Vis-NIR spectroscopy." (Page 20, Revised Manuscript).

(4). *No numbers of replicates given for the cell viability assay, or any indication what the error bars represent (Figure S12) – Actually, this rings true for all in vitro experiments and must be addressed before publication. Without replicate numbers and transparent methodology, including statistical analysis the work cannot be truly assessed.*

Response: Thank you very much for pointing this issue out, which is highly appreciated. Herein, almost all the experiments were replicated at least three times. We are sorry for this negligence and thank you very much for the kind reminding. According to the reviewer's constructive suggestion, the numbers of replicate have been supplemented for all *in vitro* and *in vivo* experiments and the manuscript has been checked thoroughly to make sure the detailed methodology transparent.

(5). *Many of the methods are not clear enough to reproduce the results. As just two examples of the many: 1) “After incubation for 24 h, the medium was discarded and exposed to H₂O₂ or H₂O₂/FeSO₄ in the presence of different concentrations from 0 to 200 µg/mL of V₂C MXenzyme for a further 12 h. – So the discarded medium was exposed to the H₂O₂ (obviously not), but the cells were? How much medium per well, what concentration of H₂O₂ was used (vital information), how many replicates. And 2) “irradiated under 365-nm UV lamp.” What manufacturer, what intensity of UV was used, at what distance from the cell etc.*

Response: We greatly appreciate the reviewer's kind reminding on the issue of detailed clarification on the experimental procedure. According to the reviewer's constructive suggestion, we have thoroughly checked the experimental section and supplemented the detailed experiment description as much as we can. For instance,

“... to verify the O₂^{•-}-scavenging ability of V₂C MXenzyme, WST-1, xanthine and XOD were mixed in V₂C dispersion with different concentrations (0, 25, 50, 100, 200 and 400 µg mL⁻¹), then... (n = 3 for each group).” (Page 19, Revised Manuscript).

“After incubation for 24 h, the old medium was discarded and the cells were exposed to fresh medium containing H₂O₂ (0, 100, 200 and 400 mM) or Fenton's reagent (400 µM of H₂O₂ and 40 µM of FeSO₄) in the presence of different concentrations of V₂C MXenzyme (0,

25, 50, 100 and 200 $\mu\text{g mL}^{-1}$) for another 12 h incubation. In order to induce the increased oxidative stress, H_2O_2 or Fenton's reagent should be freshly prepared from a stock solution before each experiment. Finally, the cell viability and ROS level were detected using CCK-8 and ROS assay, respectively ($n = 4$ for each group)." (Page 22, Revised Manuscript).

"Ultraviolet radiation-inhabitation activity. 0.8 mM of TMB solutions were placed in 1.5 mL Eppendorf tubes containing V_2C MXenzyme with different concentrations (0, 16, 32, 62.5, 125 and 250 $\mu\text{g mL}^{-1}$) and irradiation was carried out under a 365 nm ultraviolet lamp (Lichen, China) in a distance of 10 cm at a dose of 20 J cm^{-2} for 30 min. After that, the absorption changes of reaction mixture were spectrophotometrically monitored at 652 nm using UV-Vis-NIR spectroscopy ($n = 3$ for each group)." (Page 20, Revised Manuscript).

"After that, the medium was replaced with new fresh growth medium and the cells were irradiated under 365-nm UV lamp (Lichen, China) at a dose of 20 J cm^{-2} in a distance of 10 cm. After irradiation for 30 min, the cells were incubated for 4 h, and then the cell viability and ROS level were measured, respectively ($n = 5$ for each group)." (Page 22, Revised Manuscript).

"... the old medium was discarded and the cells were exposed to fresh medium containing H_2O_2 (0, 100, 200 and 400 mM) or Fenton's reagent (400 μM of H_2O_2 and 40 μM of FeSO_4) in the presence of different concentrations of V_2C MXenzyme (0, 25, 50, 100 and 200 $\mu\text{g mL}^{-1}$) for another 12 h incubation... ($n = 4$ for each group)." (Page 22, Revised Manuscript).

"... After treatment with V_2C (200 $\mu\text{g mL}^{-1}$), H_2O_2 (200 μM) and BSO (50 μM) ... ($n = 3$ for each group)" (Page 22, Revised Manuscript).

"... 500 μL of 4% paraformaldehyde ... 500 μL of 0.5% Triton X-100... 1mL of 1% BSA ... 1 mL PBS containing 1% BSA ...with 200 μL DAPI (10 $\mu\text{g mL}^{-1}$) ..." (Page 23, Revised Manuscript).

"...with 200 μL caspase-3/7 work solutions (5 μM) in PBS containing 5% FBS at 37 $^\circ\text{C}$ for at least 30 min. ... The excitation/emission maxima for the Caspase-3/7 Green Detection Reagent is 502/530 nm." (Page 23, Revised Manuscript).

(6). *Figure 6j is produced far too small for the reader to interpret any of the findings. Clear*

representative images of the model, the TH loss, through multiple brain sections are required for the reader to see the impact of the MPTP alone and with V₂C.

Response: Thank you very much for the kind suggestion, which is highly appreciated. In this revision round, we have re-conducted the experiments of *in vivo* Parkinson's disease (PD) models construction and treatment to supplement more detailed characterization data based on the reviewer's reminding and suggestion. Accordingly, clear representative immunohistochemistry and immunofluorescence images for tyrosine hydroxylase (TH), ionized calcium-binding adapter molecule 1 (IBA-1) and 4-hydroxynonenal (4-HNE) on both coronal and transverse sections of PD mice with different treatments have been supplemented in the Revised Manuscript (Fig. 6j, Page 13, Revised Manuscript) and Revised Supplementary Information (Supplementary Information, Fig. 31-37, Page 35-41, Revised Supplementary Information).

(7). The analysis of the IBA1 and 4-HNE is clearly done on different brain regions (SI Fig 21 and 22) the +V₂C group is consistently at a different anterior/posterior position and was shown to be more medial than the others. This renders the analysis meaningless. In addition to the above comment – the areas analysed are not even the injection site (stated as the striatum).

Response: Thank you very much for pointing this issue out, which is highly appreciated. According to the reviewer's comments, we have re-conducted the *in vivo* PD experiments and analyzed the TH, IBA1 and 4-HNE on almost the same mouse brain regions at striatum in the Revised Manuscript (Fig. 6j, Page 13, Revised Manuscript) and Revised Supplementary Information (Supplementary Information, Fig. 31-37, Page 35-41, Revised Supplementary Information).

(8). No injection coordinates, stereotactic information given, in sum, the PD model data cannot be repeated or properly interpreted and is not fit for publication.

Response: Thank you very much for pointing this issue out, which is highly appreciated. In this revision round, we have repeated the *in vivo* PD experiment, taken pictures and added the

injection coordinates and stereotactic information in the revised manuscript, which reads “For the stereotaxic surgery, each C57BL/6 rat was anesthetized with an intraperitoneal injection of chloral hydrate (350 mg kg⁻¹) and then placed on a Narishige stereotaxic apparatus (Yuyanbio, China) to achieve a flat skull position (Supplementary Information, Fig. 31). The scalp was incised to expose the parietal bone. After that, a small hole was drilled at the following coordinates: lateral (L) = 2.3 mm, anteroposterior (AP) = 0.8 mm, ventral (V) = -3.5 mm from the bregma point. The V₂C MXenzyme dispersion (10 mg mL⁻¹, 4 μL) or sterile saline solution (4 μL) was injected unilaterally into the striatum at the rate of 1 μL min⁻¹ and a 5 min lag was allowed before needle extraction to avoid brain damage”. (Page 25, Revised Manuscript; Page 35, Revised Supplementary Information).

(9). *I would recommend removing words such as tremendous, intriguing and admirable etc.*

Response: Thank you very much for pointing this issue out. We have deleted these words, *i.e.*, tremendous, intriguing and admirable in the revised manuscript accordingly.

(10). *The first sentence of the introduction could be broken down into two to be more clear.*

Response: Thank you very much for the suggestion. According to the reviewer’s helpful suggestion, the first sentence of the introduction have been broken down into two sentence, which reads “Reactive oxygen species (ROS) are chemically reactive molecules containing oxygen, including singlet oxygen (¹O₂), superoxide radical anion (O₂^{•-}), hydroxyl radicals (•OH) and hydrogen peroxide (H₂O₂)¹. At low concentrations, ROS play an essential role in adjusting cell functions².” (Page 2, Revised Manuscript).

(11). *...would subject biosystem to... (would subject a biosystem to).*

.... function inactively under pathological conditions... (become inactive under pathological conditions?)

found in the emerging of a large...

Despite of the high prospect...

Grammatical errors too numerous to list individually. A thorough proof read is required.

Response: Thank you very much for pointing out this grammar issue, which is highly appreciated. According to the reviewer's kind reminding, the manuscript has been polished and checked carefully and thoroughly to avoid the grammatical errors and typos, including the ones as kindly mentioned by the reviewer.

(12). *Reference required for such a statement "one of the forty essential micronutrients"*

Response: Thank you very much for the kind suggestion. We have added the reference for the statement "one of the forty essential micronutrients" in the Revised Manuscript accordingly. (Page 2, Revised Manuscript; Ref. 21 and 22).

(13). Personally, I think that Figure 1 could be greatly improved without the smileys and shield, and instead have a single image of normal cell function – highlighting every area where the V₂C can assist.

Response: Thank you very much for the suggestion. According to the reviewer's suggestion, we have improved the schematic illustration without the smileys and shield in the Revised Manuscript (Scheme 1, Page 3, Revised Manuscript) and Revised Supplementary Information (Supplementary Fig. 2 and 3, Page 5 and 6, Revised Supplementary Information).

(14). *In general a good hypothesis and clear set of objectives (or an aim) is missing from the introduction.*

Response: Thank you very much for the kind reminding, which is greatly appreciated. We have supplemented the hypothesis and clear set of objectives in the revised manuscript, which reads "Accordingly, it is necessary to develop more robust vanadium-based nanomaterials that can functionally mimic the sophisticated cellular antioxidant enzymes." (Page 2, Revised Manuscript).

"Herein, we report 2D vanadium carbide (V₂C) MXenes as a successful paradigm of guiding nanozymes to implement antioxidative behaviors for ROS elimination under pathophysiological conditions." (Page 2, Revised Manuscript).

“Our strategy not only sheds light on a novel type of nanozymes with multiple enzyme-mimicking property and excellent ROS-removal efficacy, but also paves a new avenue towards broadening the bioapplications of MXenes into catalytic nanomedicine.” (Page 3, Revised Manuscript).

(15). *Al is around 2.60 _ (units not visible in the pdf file).*

Response: Thank you very much for pointing this issue out. We have added the unit Å in the Revised Manuscript (Page 5, Revised Manuscript).

(16). *Figure 3a – Y axis label “inhibitor rate of SOD” is unclear and does not seem to match the main text which stated a drop in the formazan product. I think the aim was to mimic SOD not inhibit it.*

Response: Thank you very much for pointing this issue out, which is appreciated. We have corrected the Y axis label “Inhibitor rate of formazan formation” in Fig.3a in the Revised Manuscript (Page 7, Revised Manuscript).

(17). *Wording in the main text refers the reader to a schematic – not data “spectrophotometrically real-time measuring the decrease of NADPH level at 340 nm (Fig. 3f)”.*

Response: Thank you very much for the kind reminding. We are sincerely sorry for this mistake. We have corrected it in the revised manuscript, which reads “... spectrophotometrically real-time measuring the decrease of NADPH level at 340 nm (Fig. 3d)”. (Page 7, Revised Manuscript).

(18). *All the electron microscope characterization was carried out with using the PVA functionalized V₂C. Perhaps these studies could be included, since this is the final “product” the cells receive.*

Response: Thanks very much for the constructive suggestion. According to the reviewer’s suggestion, we herein have provided the SEM and AFM images of PVA-functionalized V₂C

in the Revised Manuscript and Revised Supplementary Information (Supplementary Fig. 19 and 20, Page 23,24, Revised Supplementary Information).

(19). No mention of rationale for choosing the cell types used.

Response: Thank you very much for the constructive and professional reminding and comment. Two different cell lines were used as evaluation systems in analyzing the apoptosis and the prevention mechanisms of antioxidants *in vitro*. The mice fibroblast cell line, L929 cells, originating from an immortalized mouse fibroblast cell line, is the internationally recognized cell line that are routinely used in *in vitro* cytotoxicity assessments (*ACS Nano* 2019, 13, 3, 3206; *Antioxidants* 2019, 8, 298; *Oxidative Med. Cell. Longev.* 2016, 2016, 3403586; *Proc. Natl. Acad. Sci. USA*, 2006, 103, 4952). The neuronal cell line, PC12 cells, derives from the neural crest that has a mixture of neuroblastic cells and eosinophilic cells, and can differentiate into neuron-like cells by nerve growth factor stimulation. PC12 cells have been widely used as a cellular model for neuronal development and neurological diseases because of their neuronal properties (*Adv. Mater.* 2016, 28, 1387; *Methods Mol. Biol.* 2012, 846, 201; *J. Neurosci.* 2002, 22, 10690; *Neurosci. Lett.* 2009, 454, 203). Accordingly, we have supplemented the rationale for choosing the cell types in the revised manuscript, which reads “Two different cell lines were used *in vitro*. The mice fibroblast cell line (L929 cells), originating from an immortalized mouse fibroblast cell line, is the internationally recognized cells that is routinely used in *in vitro* cytotoxicity assessments. In addition, the neuronal cell line (PC12 cells), derives from the neural crest that has a mixture of neuroblastic cells and eosinophilic cells, and can differentiate into neuron-like cells by nerve growth factor stimulation. PC12 cells have been widely used as a cellular model for neuronal development and neurological diseases because of their neuronal properties.” (Page 21, Revised Manuscript).

(20). “MXenzyme” line 284.

Response: Thank you very much for pointing this issue out. We have corrected it in the Revised Manuscript. (Page 9, Revised Manuscript).

(21). *Ambiguous subheadings in methods – e.g. “V₂C MXenzyme Protecting H₂O₂ L929 and PC12 Cells from Fenton’s Reagent-Induced Oxidative Stress”.*

Response: Thank you very much for pointing this issue out. According to the reviewer’s comment, we have corrected the ambiguous subheading in methods in the Revised Manuscript, which is expressed as “Protecting Cells from Fenton’s Reagent-Induced Oxidative Stress” and “Protecting Cells from UV-Induced Oxidative Stress.” (Page 22, Revised Manuscript).

REVIEWER COMMENTS

Reviewer #2 (Remarks to the Author):

The authors have made their efforts to address the reviewers questions and concerns. They have done a good job to conduct necessary experiments in order to answer the questions and thus, the quality of the work is substantially improved. I thus recommend its publication.

From Kanyi Pu at NTU

Reviewer #3 (Remarks to the Author):

The authors have addressed many of my concerns in their revised manuscript. However, I still do not find it suitable for publication without further improvement of vital aspects such as a transparent methodology and multiple sections being shown for representative images.

I am happy with the improvements made to the in vitro work, and I am unable to critically assess the ear and ankle inflammatory models as I have no experience with these.

The major issues I have with the work are as follows:

1) In their response letter, the authors state: "According to the reviewer's comments, we have re-conducted the in vivo PD experiments and analyzed the TH, IBA1 and 4-HNE on almost the same mouse brain regions at striatum in the Revised Manuscript"

I did not recommend that the study be repeated, but instead have the methodology clarified and true representative images shown.

- So why was not a more central part of the striatum analyzed (the authors state "almost the same mouse brain regions at striatum")

- How was this analysis carried out? There is absolutely no information about how TH cells were counted or whether fluorescent intensity was analyzed? (i.e. what is TH % of control on the y-axis). How many coronal sections were analyzed? Was this conducted in a blind manner? What equipment was used? How were sections mounted etc etc?

2) From how many brains was this data obtained. The methods state 6 animals per group, but the figure legend does not state that the data were obtained from these 6 animals. I only raise this as a query because the revised supplementary information shows horizontal (transverse) sections. As one can appreciate, you cannot cut the same brain in two planes for complete data! This is all very misleading/confusing, because no mention of transverse planes are mentioned in the methodology

3) What do the 3 stars placed on the MPTP group mean? Statistically significant difference from what? This should be indicated in the legend and more info added to the figure itself. If this was a difference to the control, then why was no data shown for the V2C group (or add ns where no significance is shown).

4) Previously I recommended that the authors state what the error bars represent (i.e. SEM or SD). This has not been done. These are fundamental principles of good research and will very much change the interpretation of the results by the reader, so I still think these should be included for all figures.

5) I previously recommended that the authors show multiple sections in their representative images. Although they have added a single transverse plane, they have not done this. As in point 1 above, we don't know how the quantification was carried out. Was it from a single section? Surely it was performed from multiple sections anterior to posterior, in which case the authors should lay out the representative images. For a guide to the format of this, I recommend the authors see the following publications:

Alder et al., <https://doi.org/10.1016/j.celrep.2019.08.058> (see figure 2,3 4)

Or Moloney et al., <https://doi.org/10.1016/j.brainres.2010.08.040> (see figure 2 and 3)

Although these are different stainings/methods, I have just included them to show how the data should be presented via multiple coronal sections Anterior to Posterior. This very much strengthens the readers understanding and appraisal of the work.

In my opinion, the above comments should be addressed in full before publication.

Ben Newland

Reviewer #4 (Remarks to the Author):

This is a very interesting work aimed at the development of drug delivery system for number of enzyme-mimicking compounds incorporated into vanadium carbide, V₂CMXene. The authors have appropriately addressed reviewers' comments and made significant improvements to the manuscript. However, I have several concerns regarding in vivo models for inflammation and neurodegeneration, as well as methods for assessment therapeutic efficacy of the developed formulations. Specifically, it is unclear why MPTP was administered intracranially. In contrast to 6-hydroxydopamine (6-OHDA), this toxin readily crosses the blood brain barrier and target TH-neurons in the substantia nigra (SN). Therefore, MPTP usually administered intraperitoneally. Please, explain the chosen route of toxin administration. The second concern is related to IVIS studies. Authors indicated that "fluorescent images were taken at excitation wavelength = 488 nm, and emission wavelength = 520 nm". This is wrong set up for in vivo imaging by IVIS. This type of fluorescence (wavelength) is quenched considerably by tissues and bones. Therefore, wavelength > 700 nm is usually utilized for the in vivo imaging by IVIS. The question stands, what kind of signal was recorded in the IVIS experiments and presented on Fig 6 b, f? Overall, I recommend publishing the work after these concerns will be addressed.

Response to Reviewer # 2

Comments from reviewer # 2:

The authors have made their efforts to address the reviewers questions and concerns. They have done a good job to conduct necessary experiments in order to answer the questions and thus, the quality of the work is substantially improved. I thus recommend its publication.

Response: Thanks very much for your positive comments and kind recommendation.

Response to Reviewer # 3

Comments from reviewer # 3:

The authors have addressed many of my concerns in their revised manuscript. However, I still do not find it suitable for publication without further improvement of vital aspects such as a transparent methodology and multiple sections being shown for representative images. I am happy with the improvements made to the in vitro work, and I am unable to critically assess the ear and ankle inflammatory models as I have no experience with these.

Response: Thank you very much again for careful reading of the manuscript and constructive comments that allowed us to further improve the quality of the manuscript. We agree with all your comments, and we have addressed them point by point accordingly, especially on the transparent methodology of this work and multiple sections being shown for representative images. Please find the following detailed responses to your kind comments and suggestions.

The major issues I have with the work are as follows:

1) In their response letter, the authors state: “According to the reviewer’s comments, we have re-conducted the in vivo PD experiments and analyzed the TH, IBA1 and 4-HNE on almost the same mouse brain regions at striatum in the Revised Manuscript”.

I did not recommend that the study be repeated, but instead have the methodology clarified and true representative images shown.

- So why was not a more central part of the striatum analyzed (the authors state “almost the same mouse brain regions at striatum”)

Response: Thanks very much for your comments and pointing this issue out. In last revision round, the reviewer professionally pointed out the issue on the PD experiments. Therefore, we re-conducted the experiment to make sure the whole process was right, and tried to record the details as many as possible. In order to ensure the factuality of the results, we randomly chose the brain slices which were almost the same mouse brain regions at striatum. On this basis, according to the reviewer’s comment, we have provided multiple immunohistochemical staining images of serial coronal sections including more central part of the striatum in the

Revised Manuscript and Revised Supplementary Information (Supplementary Fig.32, 35 and 38 in Page 36, 39 and 42 of Revised Supplementary Information, respectively).

2) How was this analysis carried out? There is absolutely no information about how TH cells were counted or whether fluorescent intensity was analyzed? (i.e. what is TH % of control on the y-axis). How many coronal sections were analyzed? Was this conducted in a blind manner? What equipment was used? How were sections mounted etc?

Response: We greatly appreciate the reviewer's questions. Based on the reviewer's kind reminding, we have supplemented the detailed methods in the Revised Manuscript, which reads "... The obtained eighteen brains (six per group) were fixed with 4% paraformaldehyde and embedded in paraffin. Brains in each group were randomly assigned to two subgroups: three brains were subsequently cut into 4- μ m-thick serial coronal sections and the other three brains were serially cut into 4- μ m-thick transverse sections using a pathology slicer (Leica RM2016, Germany). For immunohistochemical staining, these sections were put into xylene three times for 15 min to remove paraffin, rehydrated via a gradient of ethanol (100%, 100%, 95% and 70% ethanol) for 5 min in each concentration, and then rinsed using DI water. After that, they were brought to a boil in sodium citrate buffer (10 mM, pH 6.0) for antigen retrieval in a microwave oven (Galanz P70D20TL-P4, China), maintained at just below boiling temperature for 8 min, cooled to room temperature, placed in PBS (pH 7.4) and shaken on the decolorization shaker (Servicebio TSY-B, China) 3 times for 5 min each. Subsequently, the samples were placed in 3% H₂O₂ for 25 min and washed with PBS three times for 5 min each. To prevent antibody non-specific binding, 3% BSA solution was added to cover the samples for 30 min at room temperature. After removing the blocking solution, the sections were incubated with anti-IBA 1 antibody (1:100 dilution), anti-TH antibody (1:200 dilution) or anti-4 HNE antibody (1:100 dilution) at 4 °C overnight, and then washed by shaking 3 times for 5 min each. After covering with secondary horseradish-peroxidase-conjugated anti-rabbit IgG antibody (1:500 dilution), they were incubated for 50 min at room temperature and washed 3 times for 5 min each. Diaminobezidin (DAB) color developing solution as newly prepared was added and 5 min generally provided an acceptable staining intensity, which could be controlled under the

microscope (Nikon E100, Japan). Next, the sections were counterstained with hematoxylin for 3 min, washed with water and subsequently located in 75% ethanol for 5 min, 85% ethanol for 5 min, 100% ethanol for 5 min, 100% ethanol for 5 min, n-butanol for 5 min and xylene for 5 min. Finally, the samples were mounted onto gelatine coated microscope slides with coverslips using mounting medium, and the slides were visualized on a microscope (Nikon DS-U3, Japan).

For immunofluorescence staining, these sections were put into xylene two times for 15 min to remove paraffin, rehydrated via a gradient of ethanol (100%, 100%, 95% and 70% ethanol) for 5 min in each concentration, and then rinsed using DI water. After that, the sections were brought to a boil in ethylene diamine tetraacetic acid (EDTA) buffer (1 mM, pH 8.0) for antigen retrieval, maintained at just below boiling temperature for 8 min, cooled to room temperature and washed with PBS (pH 7.4) 3 times for 5 min each. Subsequently, the liquid was removed and the samples were marked using a hydrophobic pen. In order to prevent non-specific binding, 3% BSA solution was added to cover the marked tissue for 30 min, followed by removing the blocking solution slightly. The slides were incubated with primary antibody overnight at 4 °C and then washed with PBS (pH 7.4) 3 times for 5 min each, followed by incubation with Alexa Fluor 488 conjugated anti-rabbit IgG antibody (1:200 dilution), Alexa Fluor 594 conjugated anti-rabbit IgG antibody (1:200 dilution) and Alexa Fluor 647 conjugated anti-rabbit IgG antibody (1:200 dilution) at room temperature for 50 min in dark condition. After rinsing three times with PBS (pH 7.4) for 5 min each, the samples were treated with DAPI solution for 1 min at room temperature and washed 3 times for 5 min each. Spontaneous fluorescence quenching reagent was added and kept for 5 min. Next, the sections were mounted onto the gelatine coated microscope slides with coverslips using anti-fade mounting medium.

For the immunohistochemical staining analysis, the cell counting percentage of TH+, IBA+ or 4HNE+ cells was quantified in a blind manner, which was conducted by analyzing the treatment conditions with randomly allocated groups through the unbiased method using an image analysis program Image J (Fiji). Quantification was performed by averaging five fields per section, and five different sections were independently quantified for each mouse. Subsequently, the result was expressed as (experimental group/control group)×100%. Quantitative comparisons were carried out on image sets collected under the same imaging and

acquisition conditions.” (Page 25 and 26, Revised Manuscript).

2) *From how many brains was this data obtained. The methods state 6 animals per group, but the figure legend does not state that the data were obtained from these 6 animals. I only raise this as a query because the revised supplementary information shows horizontal (transverse) sections. As one can appreciate, you cannot cut the same brain in two planes for complete data! This is all very misleading/confusing, because no mention of transverse planes are mentioned in the methodology*

Response: Thank you very much for pointing this issue out, which is highly appreciated. According to the reviewer’s kind reminding, we have supplemented the detailed information for *in vivo* PD experiments in the revised manuscript, which reads “... The obtained eighteen brains (six per group) were fixed with 4% paraformaldehyde and embedded in paraffin. Brains in each group were randomly assigned to two subgroups: three brains were subsequently cut into 4- μ m-thick serial coronal sections and the other three brains were serially cut into 4- μ m-thick transverse sections using a pathology slicer (Leica RM2016, Germany).” (Page 25, Revised Manuscript).

3) *What do the 3 stars placed on the MPTP group mean? Statistically significant difference from what? This should be indicated in the legend and more info added to the figure itself. It this was a difference to the control, then why was no data shown for the V₂C group (or add ns where no significance is shown).*

Response: Thanks very much for your kind reminding. In the manuscript, asterisks indicate significant differences $*p < 0.05$, $**p < 0.01$ and $***p < 0.001$. According to the reviewer’s constructive suggestion, we have added the significance of test results indicated by *p* values in all the figures and legends in the Revised Manuscript, which reads

“... asterisks indicate significant difference ($*p < 0.05$, $**p < 0.01$ and $***p < 0.001$) as compared with the control group using one-way analysis of variance (ANOVA).” (Page 7, Revised Manuscript).

“...asterisks indicated significant difference ($*p < 0.05$, $**p < 0.01$ and $***p < 0.001$) as

compared with the control group using one-way analysis of variance (ANOVA).” (Page 10, Revised Manuscript).

“... and asterisks indicate significant differences ($***p < 0.001$) using one-way analysis of variance (ANOVA)” (Page 13, Revised Manuscript).

4) Previously I recommended that the authors state what the error bars represent (i.e. SEM or SD). This has not been done. These are fundamental principles of good research and will very much change the interpretation of the results by the reader, so I still think these should be included for all figures.

Response: Thank you very much for the kind suggestion, which is highly appreciated. We totally understand reviewer’s concern and we stated it in the section of “Statistical Analysis” in original manuscript: All quantitative results were shown as mean \pm standard deviation (SD). According to the reviewer’s kind suggestion, we have added replicate numbers and stated what the error bar represents in all figure legends in both Revised Manuscript and Revised Supplementary Information.

5) I previously recommended that the authors show multiple sections in their representative images. Although they have added a single transverse plane, they have not done this. As in point 1 above, we don’t know how the quantification was carried out. Was it from a single section? Surely it was performed from multiple sections anterior to posterior, in which case the authors should lay out the representative images. For a guide to the format of this, I recommend the authors see the following publications: Alder et al., <https://doi.org/10.1016/j.celrep.2019.08.058> (see figure 2,3 4) Or Moloney et al., <https://doi.org/10.1016/j.brainres.2010.08.040> (see figure 2 and 3). Although these are different stainings/methods, I have just included them to show how the data should be presented via multiple coronal sections Anterior to Posterior. This very much strengthens the readers understanding and appraisal of the work. In my opinion, the above comments should be addressed in full before publication.

Response: We appreciate very much for your professional comments and suggestion. Based on

the reviewer's constructive suggestion, we have provided multiple immunohistochemical staining images of serial coronal sections in both Revised Manuscript and Revised Supplementary Information to strengthen the readers understanding and appraisal of this work, which are shown in Supplementary Fig.32, 35 and 38. (Page 36, 39 and 42, Revised Supplementary Information).

Finally thank you very much for your professional comments and constructive suggestions, which are greatly appreciated and highly helpful for further improving the manuscript quality. We sincerely hope that the revised manuscript has addressed all your comments and suggestions.

Response to Reviewer # 4

Comments from reviewer #4:

This is a very interesting work aimed at the development of drug delivery system for number of enzyme-mimicking compounds incorporated into vanadium carbide, V₂CMXene. The authors have appropriately addressed reviewers' comments and made significant improvements to the manuscript. However, I have several concerns regarding in vivo models for inflammation and neurodegeneration, as well as methods for assessment therapeutic efficacy of the developed formulations.

Response: Thanks very much for your positive comments and kind concerns. Please find the following detailed responses to your comments and concerns.

1) Specifically, it is unclear why MPTP was administered intracranially. In contrast to 6-hydroxydopamine (6-OHDA), this toxin readily crosses the blood brain barrier and target TH-neurons in the substantia nigra (SN). Therefore, MPTP usually administered intraperitoneally. Please, explain the chosen route of toxin administration.

Response: Thanks very much for your professional suggestion. We totally understand the reviewer's concern and we are sorry for this misunderstanding. MPTP as a lipophilic compound can cross the blood-brain barrier and we administered it intraperitoneally in this work. In the section of "Parkinson's Disease (PD) Models Construction and Treatment" (Page 25, Original Manuscript), we described "eighteen C57BL/6 mice were randomly assigned and treated with 30 mg kg⁻¹ MPTP saline solution intraperitoneally once daily for 10 consecutive days, along with daily saline-treated as control". According to the reviewer's comment and in order to avoid readers' misunderstanding, besides in the section of "Methods", we have emphasized it in the section of "Results and discussion" in Revised Manuscript, which reads "MPTP, as a lipophilic compound, can cross the blood-brain barrier, which was administered intraperitoneally." (Page 14, Revised Manuscript).

2) The second concern is related to IVIS studies. Authors indicated that "fluorescent images

were taken at excitation $wl = 488\text{ nm}$, and emission $wl = 520\text{ nm}$ ". This is wrong set up for in vivo imaging by IVIS. This type of fluorescence (wl) is quenched considerably by tissues and bones. Therefore, $wl > 700\text{ nm}$ is usually utilized for the in vivo imaging by IVIS. The question stands, what kind of signal was recorded in the IVIS experiments and presented on Fig 6 b, f? Overall, I recommend publishing the work after these concerns will be addressed.

Response: Thanks very much for your kind comments, which are highly appreciated. Based on the reviewer's concern, we have further checked the device configuration of IVIS Lumina Series III imaging system and consulted the engineer from PerkinElmer company. In this IVIS imaging system, we equipped with up to 26 filter sets that can be used to image reporters, which emit from green to near-infrared light including 19 excitation filters of 420, 440, 460, 480, 500, 520, 540, 560, 580, 600, 620, 640, 660, 680, 700, 720, 740, 760 and 780 nm, and 7 emission filters of 520, 570, 620, 670, 710, 790 and 845 nm. These parameters meet our research requirements. Furthermore, multispectral unmixing with Compute Pure Spectrum technology is used to accurately differentiate 2',7'-dichlorofluorescein (DCF) signal from autofluorescence. As mentioned by the reviewer, the fluorescence of DCF is quenched considerably by hair, tissues and bones. In this work, according to the published literatures (*Chem* 2019, 5, 2378; *Chem. Sci.* 2018, 9, 2927; *ACS Nano* 2018, 12, 8882; *Angew. Chem. Int. Ed.* 2016, 55, 6646), we removed all the mice hair, and then the models of inflammation were established at the ear and ankle, in which the skin is thin and surround on both sides by epithelia without much muscle or bone disturbances. Additionally, we also examined H&E-stained histological sections of these samples for further evaluation. Therefore, according to the reviewer's suggestion, we have added the device configuration of IVIS imaging system in the Revised Manuscript, which reads "..., the whole-body photoluminescence imaging was acquired on a PerkinElmer IVIS Lumina Series III imaging system (the excitation filters of 420, 440, 460, 480, 500, 520, 540, 560, 580, 600, 620, 640, 660, 680, 700, 720, 740, 760 and 780 nm, and the emission filters of 520, 570, 620, 670, 710, 790 and 845 nm) using an excitation wavelength of 480 nm and an emission wavelength of 520 nm...Furthermore, multispectral unmixing with Compute Pure Spectrum technology was used to enable accurate autofluorescence removal." (Page 25, Revised Manuscript).

Finally thank you very much for your professional comments and constructive suggestions, which are greatly appreciated and highly helpful for further improving the manuscript quality. We sincerely hope that the revised manuscript has addressed all your comments and suggestions.

REVIEWERS' COMMENTS

Reviewer #3 (Remarks to the Author):

The authors have addressed my concerns about the transparency of the methods, and have added in representative images in the supplementary information which are useful for assessing the extent of TH neuron preservation. I therefore no longer have any queries about this side of the work (n.b. I have no expertise in the ear or ankle model). I can therefore recommend this work for publication. Two minor notes – the authors have misspelt diaminobenzidine, and I recommend that the authors state the source and catalogue number for the antibodies used.

I hope my comments have served to improve the quality of this publication, and have highlighted the methodological details that are required. Thanks you for addressing them in full.

Regards,

Ben Newland

Reviewer #4 (Remarks to the Author):

All my concerns were addressed in the revised version. Thank you.

Response to Reviewer # 3

Comments from reviewer # 3:

The authors have addressed my concerns about the transparency of the methods, and have added in representative images in the supplementary information which are useful for assessing the extent of TH neuron preservation. I therefore no longer have any queries about this side of the work (n.b. I have no expertise in the ear or ankle model). I can therefore recommend this work for publication. Two minor notes – the authors have misspelt diaminobenzidine, and I recommend that the authors state the source and catalogue number for the antibodies used. I hope my comments have served to improve the quality of this publication, and have highlighted the methodological details that are required. Thank you for addressing them in full.

Response: Thanks very much for your positive comments and kind recommendation. Thank you for pointing out the issues concerning the spelling mistakes and the source and catalogue number for the antibodies. Based on the reviewer's kind reminding, we have corrected the misspelt diaminobenzidine and supplemented the source and catalogue number for the antibodies in the Revised Manuscript, which reads:

“... Diaminobenzidine (DAB) color developing solution...” (Page 27, Revised Manuscript)

“... Caspase-3 antibody (CST #9662), cleaved caspase-3 (CST #9661) and glyceraldehyde-3-phosphate dehydrogenase (GAPDH, CST #2118) antibody were purchased from Cell Signaling Technology Inc. (Danvers, Massachusetts, USA). Anti-tyrosine hydroxylase (abcam #ab112), anti-4 hydroxynonenal (abcam #ab46545), anti-IBA 1 (abcam #178847) antibody were obtained from Abcam Inc. (Cambridge, Massachusetts, USA).” (Page 19, Revised Manuscript)

Finally, thank you very much again for careful reading of the manuscript and constructive comments that allowed us to further improve the quality of the manuscript.

Response to Reviewer # 4

Comments from reviewer # 4:

All my concerns were addressed in the revised version. Thank you.

Response: Thanks very much for your positive comments.